

# Sulfur dioxide retrievals from TROPOMI onboard Sentinel-5 Precursor: Algorithm Theoretical Basis

N. Theys[1], I. De Smedt[1], H. Yu[1], T. Danckaert[1], J. van Gent[1], C. Hörmann[2], T. Wagner[2], P. Hedelt[3], H. Bauer[3], F. Romahn[3], M. Pedergnana[3], D. Loyola[3], M. Van Roozendael[1]

[1]{Royal Belgian Institute for Space Aeronomy (BIRA-IASB), Brussels, Belgium}

[2]{Max Planck Institute for Chemistry (MPIC), Hahn-Meitner-Weg 1, 55128 Mainz, Germany}

[3]{Institut für Methodik der Fernerkundung (IMF), Deutsches Zentrum für Luft und Raumfahrt (DLR), Oberpfaffenhofen, Germany}

Correspondence to N. Theys (theys@aeronomie.be)

**ABSTRACT**

The TROPOspheric Monitoring Instrument (TROPOMI) onboard the Copernicus Sentinel-5 Precursor (S-5P) platform will measure ultraviolet Earthshine radiances at high spectral and improved spatial resolution (pixel size of 7x3.5 km² at nadir) compared to its predecessors OMI and GOME-2. This paper presents the sulfur dioxide ($SO_2$) vertical column retrieval algorithm implemented in the S-5P operational processor UPAS (Universal Processor for UV/VIS Atmospheric Spectrometers), and comprehensively describes its various retrieval steps. The spectral fitting is performed using the Differential Optical Absorption Spectroscopy (DOAS) method including multiple fitting windows to cope with the large range of atmospheric $SO_2$ columns encountered. It is followed by a slant column background correction scheme to reduce possible biases or across-track dependent artifacts in the data. The $SO_2$ vertical columns are obtained by applying Air Mass Factors (AMF) calculated for a set of representative a-priori profiles and accounting for various parameters influencing the retrieval sensitivity to $SO_2$. Finally, the algorithm includes an error analysis module which is fully described here. We also discuss verification results (as part of the algorithm development) and future validation needs of the TROPOMI $SO_2$ algorithm.





## 1.  INTRODUCTION

Sulfur dioxide enters the Earth's atmosphere via both natural and anthropogenic processes.
Through the formation of sulfate aerosols and sulfuric acid, it plays an important role on the
chemistry at local and global scales and its impact ranges from short term pollution to
climate forcing. While about one third of the global sulfur emissions originates from natural
sources (volcanoes and biogenic dimethyl sulfide), the main contributor to the total budget
is from anthropogenic emissions mainly from the combustion of fossil fuels (coal and oil) and
from smelting. Over the last decades, a host of satellite-based UV-visible instruments have
been used for the monitoring of anthropogenic and volcanic $SO_2$ emissions. Total vertical
column density (VCD) of $SO_2$ has been retrieved with the sensors TOMS (Krueger, 1983),
GOME (Eisinger and Burrows, 1998; Thomas et al., 2005; Khokar et al., 2005), SCIAMACHY
(Afe et al., 2004), OMI (Krotkov et al., 2006; Yang et al., 2007, 2010; Li et al., 2013; Theys et
al., 2015), GOME-2 (Richter et al., 2009; Bobrowski et al., 2010; Nowlan et al., 2011; Rix et
al., 2012; Hörmann et al., 2013) and OMPS (Yang et al., 2013). In particular, the Ozone
Monitoring Instrument (OMI) has largely demonstrated the value of satellite UV-visible
remote-sensing (1) in monitoring volcanic plumes in near-real time (Brenot et al., 2014) and
changes in volcanic degassing at the global scale (Carn et al., 2016, and references therein),
(2) in detecting and quantifying large anthropogenic $SO_2$ emissions, weak or unreported
emission sources worldwide (Theys et al., 2015; Fioletov et al., 2016; McLinden et al., 2016)
as well as investigating their long-term changes (Krotkov et al., 2016; van der A et al., 2016).
An exemplary map of OMI $SO_2$ columns (Theys et al., 2015) averaged over the 2005-2009
period is shown in Figure 1, illustrating typical anthropogenic emission hotspots (China,
Eastern Europe, India and the Middle East) and signals from volcanic activity (e.g. from the
volcanoes in D.R. Congo).
The 7-year lifetime Sentinel-5p sensor TROPOMI (Veefkind et al., 2012) will fly on a polar low
earth orbit with a wide swath of 2600 km. The TROPOMI instrument is a push-broom
imaging spectrometer similar in concept as OMI. It has eight spectral bands covering UV to
SWIR wavelengths. The $SO_2$ retrieval algorithm exploits measurements from band 3 (310-405
nm), with typical spectral resolution of 0.54 nm, signal-to-noise ratio of about 1000 and pixel
size as good as 7x3.5 km².



TROPOMI will continue and improve the measurement time-series of OMI SO$_2$ and other UV
sensors. Owing to similar performance as OMI in terms of signal-to-noise ratio and
unprecedented spatial resolution, TROPOMI will arguably discern very fine details in the SO$_2$
distribution and will be able to detect point sources with annual SO$_2$ emissions of about 10
kT/year or lower (using oversampling techniques).
This paper gives a thorough description of the operational TROPOMI SO$_2$ algorithm and
reflects the S5P SO$_2$ L2 Algorithm Theoretical Basis Document v1.0. In Section 2, we first
present the product requirements and briefly discuss the expected product performance in
terms of precision and accuracy. It is then followed by the SO$_2$ column retrieval algorithm
description. An error analysis of the retrieval method is presented in Section 3. Results from
algorithm verification exercise using an independent retrieval scheme is given in Section 4.
The possibilities for future validation of the retrieved SO$_2$ data product can be found in
Section 5. Conclusions are given in Section 6. Additional information on data product and
auxiliary data are provided in annex.
**2.  TROPOMI SO$_2$ ALGORITHM**
**2.1 PRODUCT REQUIREMENTS**
While UV measurements are highly sensitive to SO$_2$ at high altitudes (upper troposphere-
lower stratosphere), the sensitivity to SO$_2$ concentration in the boundary layer is intrinsically
limited from space due to the combined effect of scattering (Rayleigh and Mie) and ozone
absorption that hamper the penetration of solar radiation into the lowest atmospheric
layers. Furthermore the SO$_2$ absorption signature suffers from the interference with the
ozone absorption spectrum.
The retrieval precision (or random uncertainty) is driven by the signal to noise ratio of the
recorded spectra and by the retrieval wavelength interval used, the accuracy (or systematic
uncertainty) is limited by the knowledge on the auxiliary parameters needed in the different
retrieval steps. Among these are the treatment of other chemical interfering species, clouds
and aerosol, the representation of vertical profiles (gas, temperature, pressure), and
uncertainties on data from external sources (e.g., surface reflectance).



Requirements on the accuracy and precision for the data products derived from the
TROPOMI measurements are specified in the GMES Sentinels 4 and 5 and 5p Mission
Requirements Document MRD (Langen et al., 2011), the Report of The Review Of User
Requirements for Sentinels-4/5 (Bovensmann et al., 2011) and the Science Requirements
Document for TROPOMI (van Weele et al., 2008). These requirements derive from the
CAPACITY study (Kelder et al., 2005) and have been fine-tuned by the CAMELOT (Levelt et
al., 2009) and ONTRAQ (Zweers et al., 2010) studies. The CAPACITY study has defined three
main themes: The ozone layer (A), air quality (B), and climate (C) with further division into
sub themes. Requirements for $SO_2$ have been specified for a number of these sub themes. In
the following paragraphs, we discuss these requirements and the expected performances of
the $SO_2$ retrieval algorithm (summary is given in Table 1).
*Theme A3 - Ozone layer assessment*
This theme addresses the importance of measurements in the case of enhanced $SO_2$
concentrations in the stratosphere due to severe volcanic events. Long-term presence (up to
several months) of $SO_2$ in the stratosphere contributes to the stratospheric aerosol loading
and hence affects the climate and the stratospheric ozone budget. For such scenarios, the
requirements state that the stratospheric vertical column should be monitored with a total
uncertainty of 30%. Although powerful volcanic events generally produce large amounts of
$SO_2$, monitoring such a plume over extended periods of time requires the detection of the
plume also after it has diluted during the weeks after the eruption.



From an error analysis of the proposed $SO_2$ algorithm (Section 3), we have assessed the
major sources of uncertainty in the retrieved $SO_2$ column. One of the main contributors to
the total uncertainty is instrumental noise. This source of error alone limits the precision to
vertical columns above about 0.25 DU (1 DU=2.69 x $10^{16}$ molec.cm$^{-2}$). For $SO_2$ in the
stratosphere, the summing up of the various uncertainties (Section 3) is believed to be
around the required uncertainty of 30% for diluted $SO_2$ plumes, provided that the vertical
column is larger than 0.5 DU. Explosive volcanic eruptions capable of injecting $SO_2$ into the
stratosphere regularly show stratospheric $SO_2$ columns of a few DU to several hundreds of
DU or more, as was the case, for example, for the eruptions of Mt. Kasatochi (Yang et al.,
2010) and Sarychev Peak (Carn et al., 2011). For very large $SO_2$ concentrations, the
dynamical use of different fitting windows (see section 2.2) enables to reach 30 %
uncertainty level.
*Theme B – Air quality*
This theme includes three sub themes:
B1 -Protocol monitoring: This involves the monitoring of abundances and concentrations
of atmospheric constituents, driven by several agreements, such as the Gothenburg
protocol, National Emission Ceilings, and EU Air Quality regulations.
B2 -Near-real time (NRT) data requirements: This comprises the relatively fast (~30
minutes) prediction and determination of surface concentrations in relation to health
and safety warnings.
B3 – Assessment: This sub theme aims at answering several air quality related scientific
questions, such as the effect on air quality of special and temporal variations in oxidizing
capacity and long-range transport of atmospheric constituents.
A more detailed description of the air quality sub themes can be found in Langen et al.

26  (2011).



The user requirements on $SO_2$ products are equal for all three sub themes. For the total
vertical column and the tropospheric vertical column of $SO_2$, the user requirements state an
absolute maximum uncertainty of 1.3 x $10^{15}$ molecules cm$^{-2}$ or 0.05 DU. This number derives
from the ESA CAPACITY study, where the number was expressed as 0.4 ppbv for a 1.5 km
thick boundary layer reaching up to 850 hPa. From the uncertainty due to instrument noise
only, it is clear that the 0.05 DU requirement cannot be met on a single-measurement basis.
This limitation was already found in the ESA CAMELOT study (Levelt et al., 2009).
For anthropogenic $SO_2$ typically confined in the planetary boundary layer (PBL), calculations
performed within the CAMELOT study showed that the smallest vertical column that can be
detected in the PBL is of about 1-3 DU (for a signal-to-noise ratio (SNR) of 1000). Although
pollution hotspots can be better identified by spatial or temporal averaging, several
uncertainties (e.g. due to varying surface albedo or $SO_2$ vertical profile shape) are not
averaging out and directly limit the product accuracy to about 50% or more. Though the
difference between the MRD requirements and the expected TROPOMI performance is
rather large, one could argue that the required threshold should not be a strict criterion in all
circumstances. The user requirement of 0.05 DU represents the maximum uncertainty to
distinguish (anthropogenic) pollution sources from background concentrations. Bovensmann
et al. (2011) reviewed the MRD user requirements and motivated a relaxation of certain user
requirements for specific conditions. For measurements in the PBL, the document proposes
a relative requirement of 30-60% in order to discriminate between enhanced (> 1.5 ppbv),
moderate (0.5-1.5 ppbv), and background concentrations (<0.5 ppbv). It is expected that it
will be possible to discriminate these three levels by averaging (spatially-temporally)
TROPOMI data.
For volcanic $SO_2$ plumes in the free-troposphere, a better measurement sensitivity is
expected for TROPOMI. The expected precision is about 0.5 DU on the vertical column. The
accuracy on the $SO_2$ vertical column will be strongly limited by the $SO_2$ plume height and the
cloud conditions. As these parameters are highly variable in practice, it is difficult to
ascertain the product accuracy for these conditions.





**2.2 ALGORITHM DESCRIPTION**
The first algorithm to retrieve $SO_2$ columns from space-borne UV measurements was
developed based on a few wavelength pairs (for TOMS) and has been subsequently applied
and refined for OMI measurements (e.g., Krotkov et al., 2006; Yang et al., 2007 and
references therein). Current algorithms exploit back-scattered radiance measurements in a
wide spectral range using a direct fitting approach (Yang et al., 2010; Nowlan et al., 2011), a
Principal Component Analysis (PCA) method (Li et al., 2013) or (some form of) Differential
Optical Absorption Spectroscopy (DOAS; Platt and Stutz, 2008), see e.g. Richter et al. (2009),
Hörmann et al. (2013), Theys et al. (2015).
Direct fitting schemes in which on-the-fly radiative transfer simulations are made for all
concerned wavelengths and resulting simulated spectra are adjusted to the spectral
observations, are in principle the most accurate. They are able to cope with very large $SO_2$
columns (such as those occurring during explosive volcanic eruptions), i.e. conditions
typically leading to a strongly non-linear relation between the $SO_2$ signal and the VCD.
However, the main disadvantage of direct fitting algorithms with respect to DOAS (or PCA), is
that they are computationally expensive and are out of reach for TROPOMI operational near-
real-time processing, for which the Level 1b data flow is expected to be massive and deliver
around 1,5 million spectral measurements per orbit (~15 orbits daily) for band 3 (with a
corresponding data size of 6 gigabytes). To reach the product accuracy and processing
performance requirements, the here adopted approach applies DOAS in three different
fitting windows (within the 310-390 nm spectral range) that are still sensitive enough to $SO_2$
but less affected by non-linear effects (Bobrowski et al., 2010; Hörmann et al., 2013).


Figure 2 shows the full flow diagram of the SO$_2$ retrieval algorithm including the
dependencies on auxiliary data and other L2 products. The algorithm and its application to
OMI data is also described  in Theys et al. (2015), although there are differences in some
settings. The baseline operation flow of the scheme is based on a DOAS retrieval algorithm
and is identical to that implemented in the retrieval algorithm for HCHO (also developed by
BIRA-IASB, see De Smedt et al., 2016). The main output parameters of the algorithm are SO$_2$
vertical column density, slant column density, air mass factor, averaging kernels (AK) and
error estimates. Here, we will first briefly discuss the principle of the DOAS VCD retrieval
before discussing the individual steps of the process in more details.
First, the radiance and irradiance data are read from a S5P L1b file, along with geolocation
data such as pixel coordinates and observation geometry (sun and viewing angles). At this
stage also cloud cover information is obtained from the S5P cloud L2 data, as required for
the calculation of the AMF, later in the scheme. Then relevant absorption cross section data,
as well as characteristics of the instrument (e.g., slit functions) are used as input for the SO$_2$
slant column density determination. As a baseline, the slant column fit is done in a sensitive
window from 312 to 326 nm. For pixels with a strong SO$_2$ signal, results from alternative
windows, where the SO$_2$ absorption is weaker can be used instead. An empirical offset
correction (dependent on the fitting window used) is then applied to the SCD. The latter
correction accounts for systematic biases in the SCDs. Following the SCD determination, the
AMF is estimated based on a pre-calculated weighting functions (or box-AMFs) look-up table
(LUT). This look-up-table is generated using the LInearized Discrete Ordinate Radiative
Transfer (LIDORT) code (Spurr, 2008) and has several entries: cloud cover data, topographic
information, observation geometry, surface albedo, effective wavelength (representative of
the fitting window used), total ozone column and the shape of the vertical SO$_2$ profile. The
algorithm also includes an error calculation and retrieval characterization module (Section 3)
that computes the averaging kernels (Eskes & Boersma, 2003), which characterize the
vertical sensitivity of the measurement and which are required for comparison with other
types of data (Veefkind et al., 2012).
The final SO$_2$ vertical column is obtained by:
$N_v = \dfrac{N_S - N_S^{back}}{M}$                                                (1)



where the main quantities are the vertical column ($N_v$), the slant column density ($N_s$) and the
values used for the background correction ($N_s^{back}$). M is the air mass factor.
**2.2.1   Slant column retrieval**
The backscattered radiance spectrum recorded by the space instrument differs from the
solar spectrum because of the interactions of the photons with the Earth's atmosphere and
surface reflection. Hence the reflectance spectra contains spectral features that can be
related to the various absorbing species and their amounts in the atmosphere. The DOAS
method aims at the separation of the highly structured trace gas absorption spectra and
broadband spectral structures. The technique relies on a number of assumptions that can be
summarized as follows:
a.  The spectral analysis and atmospheric radiative transfer computations are treated
separately, by considering one averaged atmospheric light path of the photons
travelling from the sun to the instrument.
b.  The absorption cross-sections are not strongly dependent on pressure and
temperature. Additionally, the averaged light path should be weakly dependent on
the wavelength - for the fitting window used - which enables to define an effective
absorption (slant) column density. It should be noted that strictly this is not valid for
the $SO_2$ DOAS retrieval because of strong absorption by ozone and in some cases  $SO_2$
itself (for large $SO_2$ amounts).
c.  Spectrally smoothed structures due broadband absorption, scattering and reflection
processes can be well reproduced by a low-order polynomial as a function of
wavelength.
Photons collected by the satellite instrument may have followed very different light paths
through the atmosphere depending on their scattering history. However, a single effective
light path is assumed, which represents an average of the complex paths of all reflected and
scattered solar photons reaching the instrument within the spectral interval used for the
retrieval. This simplification is valid if the effective light path is reasonably constant over the
considered wavelength range. The spectral analysis can be described by the following
equation:



$$\ln \frac{\pi I(\lambda)}{\mu_0 E_0(\lambda)} = -\sum_j \sigma_j(\lambda)\, Ns_j + \sum_p c_p \lambda^p \tag{2}$$

Here, $I(\lambda)$ is the observed backscattered Earthshine radiance [W m$^{-2}$nm$^{-1}$sr$^{-1}$], $E_0$ is the solar
irradiance [W m$^{-2}$nm$^{-1}$] and $\mu_0 = \cos\theta_0$. The first term on the right hand side indicates all
relevant absorbing species with absorption cross-sections $\sigma_j$ [cm$^2$ molec.$^{-1}$]. Integration of
the number densities of these species along the effective light path gives the slant column
density $Ns_j$ [molec.cm$^{-2}$]. Equation 2 can be solved by least-squares fitting techniques (Platt
and Stutz, 2008) for the slant column values. The final term in Eq. 2 is the polynomial
representing broad band absorption and (Rayleigh and Mie) scattering structures in the
observed spectrum and also accounts for possible errors such as e.g. uncorrected instrument
degradation effects, uncertainties in the radiometric calibration or possible residual
(smooth) polarization response effects not accounted for in the level 0-1 processing.
Apart from the cross-sections for the trace gases of interest, additional fit parameters need
to be introduced to account for the effect of several physical phenomena on the fit result.
For $SO_2$ fitting, these are the filling-in of Fraunhofer lines (Ring effect) and the need for an
intensity offset-correction. In the above, we have assumed that for the ensemble of
observed photons a single effective light path can be assumed over the adopted wavelength
fitting interval. For the observation of (generally small) $SO_2$ concentrations at large solar
zenith angles (SZA) this is not necessarily the case. For such long light paths, the large
contribution of $O_3$ absorption may lead to negative $SO_2$ retrievals. This may be mitigated by
taking the wavelength dependence of the $O_3$ SCD over the fitting window into account, as
will be described in the next section.
The different parts of the DOAS retrieval are detailed in the next subsections and Table 2
gives a summary of settings used to invert $SO_2$ slant columns. Note that in Eq. 2, the daily
solar irradiance is used as a baseline for the reference spectrum. As a better option, it is
generally preferred to use daily averaged radiances, selected for each across-track position,
in the equatorial Pacific. In the NRT algorithm, the last valid day can be used to derive the
reference spectra, while in the offline version of the algorithm, the current day should be
used. Based on OMI experience, it would allow e.g. for better handling of instrumental



artifacts and degradation of the recorded spectra for each detector. At the time of writing, it
is planned to test this option during the S5P commissioning phase.
2.2.1.1 Wavelength fitting windows
DOAS measurements are in principle applicable to all gases having suitable narrow
absorption bands in the UV, visible, or near IR regions. However, the generally low
concentrations of these compounds in the atmosphere, and the limited signal-to-noise ratio
of the spectrometers, restrict the number of trace gases that can be detected. Many spectral
regions contain several interfering absorbers and correlations between absorber cross-
sections can sometimes lead to systematic biases in the retrieved slant columns. In general,
the correlation between cross-sections decreases if the wavelength interval is extended, but
then the assumption of a single effective light path defined for the entire wavelength
interval may not be fully satisfied, leading to systematic misfit effects that may also
introduce biases in the retrieved slant columns (e.g., Pukīṭe et al., 2010) . To optimize DOAS
retrieval settings, a trade-off has to be found between these effects. In the UV-visible
spectral region, the cross-section spectrum of $SO_2$ has its strongest bands in the 280-320 nm
range (Figure 3). For the short wavelengths in this range, the $SO_2$ signal however suffers
from a strong increase in Rayleigh scattering and ozone absorption. In practice, this leads to
a very small $SO_2$ signal in the satellite spectra compared to ozone absorption, especially for
tropospheric $SO_2$. Consequently, $SO_2$ is traditionally retrieved (for GOME, SCIAMACHY,
GOME-2, OMI) using sensitive windows in the 310-326 nm range. Note that even in this
range the $SO_2$ absorption can be three orders of magnitude lower than that of ozone.
The TROPOMI $SO_2$ algorithm is using a multiple windows approach:
• 312-326 nm: classical fitting window, ideal for small columns. This window is used as

25       baseline. If non-linear effects due to high $SO_2$ amounts are encountered, one of the

26       two following windows will be used instead.

• 325-335 nm: in this window, differential $SO_2$ spectral features are one order of

28       magnitude smaller than in the classical window. It allows the retrieval of moderate

29       $SO_2$ columns, an approach similar to the one described by Hörmann et al. (2013).



• 360-390 nm: SO$_2$ absorption bands are 2-3 orders of magnitude weaker than in the

2        classical window and are best suited for the retrieval of extremely high SO$_2$ columns

3        (Bobrowski et al., 2010)

Note that in the 325-335 nm and 360-390 nm windows the Rayleigh scattering and ozone
absorption are less important than in the baseline 312-326 nm window (see also Figure 3).
Specifically, in the first two intervals, absorption cross-sections of O$_3$ at 228K and 243K are
included in the fit and, to better cope with the strong (non-linear) ozone absorption at short
wavelengths, the retrieval also includes two pseudo cross-sections following the approach of
Puķīte et al. (2010): $\lambda\sigma_{O3}$ and $\sigma_{O3}^2$ calculated from the O$_3$ cross-section spectrum at 228K.
The correction for the Ring effect is based on the technique outlined by Vountas et al.
(1998). This technique involves a Principal Component Analysis of a set of Ring spectra,
calculated for a range of solar zenith angles. The first two of the resulting eigenvectors
appear to accurately describe the Ring spectra, with the first eigenvector representing the
filling-in of Fraunhofer lines and the second mostly representing the filling-in of gas
absorption features. In the retrieval algorithm, these vectors are determined by
orthogonalizing two Ring spectra, calculated by LIDORT-RRS (Spurr et al., 2008), a version of
LIDORT accounting for rotational Raman scattering, for a low SZA (20°) and a high SZA (87°),
respectively.
2.2.1.2 Wavelength calibration and convolution to TROPOMI resolution
The quality of a DOAS fit critically depends on the accuracy of the alignment between the
earthshine radiance spectrum, the reference spectrum and the cross-sections. Although the
Level 1b will contain a spectral assignment, an additional spectral calibration is part of the
SO$_2$ algorithm. Moreover, the DOAS spectral analysis includes also the fit of shift and stretch
of radiance spectra because the TROPOMI spectral registration will differ from one ground-
pixel to another e.g. due to thermal variations over the orbit as well as due to
inhomogeneous filling of the slit in flight direction.
The wavelength registration of the reference spectrum can be fine-tuned by means of a
calibration procedure making use of the solar Fraunhofer lines. To this end, a reference solar
atlas $E_s$ accurate in absolute vacuum wavelength to better than 0.001 nm (Chance and



Kurucz, 2010) is degraded at the resolution of the instrument, through convolution by the
TROPOMI instrumental slit function.
Using a non-linear least-squares approach, the shift ($\Delta_i$) between the reference solar atlas
and the TROPOMI irradiance is determined in a set of equally spaced sub-intervals covering a
spectral range large enough to encompass all relevant fitting intervals. The shift is derived
according to the following equation:

$$E_0(\lambda) = E_s(\lambda - \Delta_i) \tag{3}$$

where $E_s$ is the solar spectrum convolved at the resolution of the instrument and $\Delta_i$ is the
shift in sub-interval $i$. A polynomial is then fitted through the individual points in order to
reconstruct an accurate wavelength calibration $\Delta(\lambda)$ for the complete analysis interval. Note
that this approach allows to compensate for stretch and shift errors in the original
wavelength assignment.
In the case of TROPOMI, the procedure is complicated by the fact that such calibrations must
be performed (and stored) for each separate spectral field on the CCD detector array. Indeed
due to the imperfect characteristics of the imaging optics, each row of the TROPOMI
instrument must be considered as a separate spectrometer for analysis purposes.
In a subsequent step of the processing, the absorption cross-sections of the different trace
gases must be convolved with the instrumental slit function. The baseline approach is to use
slit functions determined as part of the TROPOMI key data. Slit functions are delivered for
each binned spectrum and as a function of wavelength. Note that an additional feature of
the prototype algorithm allows to dynamically fit for an effective slit function of known line
shape (e.g. asymmetric Gaussian). This can be used for verification and monitoring purpose
during commissioning and later on during the mission.
More specifically, wavelength calibrations are made for each TROPOMI orbit as follows:
1.  The TROPOMI irradiances (one for each row of the CCD) are calibrated in wavelength

25       over the 310-390 nm wavelength range, using 10 sub-windows.

2.  The earthshine radiances and the absorption cross-sections are interpolated (cubic

27       spline interpolation) on the calibrated wavelength grid, prior to the analysis.



3.  During spectral fitting, shift and stretch parameters are further derived to align

2        radiance and irradiance spectra. The reference wavelength grid used in the DOAS

3        procedure is the (optimized) grid of the TROPOMI solar irradiance.

2.2.1.3 Spike removal algorithm
A method to remove individual hot pixels or pixels affected by the South Atlantic Anomaly
has been presented for $NO_2$ retrievals in Richter et al. (2011). Often only a few individual
detector pixels are affected and in these cases, it is possible to identify and remove the noisy
points from the fit. However, as the amplitude of the distortion is usually only of the order of
a few percent or less, it cannot always be found in the highly structured spectra themselves.
Higher sensitivity for spikes can be achieved by analysing the residual of the fit where the
contribution of the Fraunhofer lines, scattering, and absorption is already removed.
When the residual for a single pixel exceeds the average residual of all pixels by a chosen
threshold ratio (the tolerance factor), the pixel is excluded from the analysis, in an iterative
process. This procedure is repeated until no further outliers are identified, or until the
maximum number of iterations is reached (here fixed to 3). This is especially important to
handle the degradation of 2-D detector arrays such as OMI or TROPOMI. However, this
improvement of the algorithm has a non-negligible impact on the time of processing.  At the
time of writing, the exact values for the tolerance factor and maximum number of iterations
of the spike removal procedure are difficult to ascertain and will only be known during
operations. To assess the impact on the processing time, test retrievals have been done on
OMI spectra using a tolerance factor of 5, and a limit of 3 iterations (this could be relaxed)
and it leads to an increase in processing time by a factor of 1.5.





2.2.1.4 Fitting window selection
The implementation of the multiple fitting windows retrieval requires selection criteria for
the transition from one window to another. These criteria are based on the measured $SO_2$
slant columns. As a baseline, the $SO_2$ SCD in the 312-326 nm window will be retrieved for
each satellite pixel. When the resulting value exceeds a certain criterion, the slant column
retrieval is taken from an alternative window. As part of the algorithm development and
during the verification exercise (Section 4), closed-loop retrievals have been performed and
application of the algorithm to real data from the GOME-2 and OMI instruments lead to
threshold values and criteria as given in Table 3.
**2.2.2    Offset correction**
When applying the algorithm to OMI and GOME-2 data, across-track/viewing angle
dependent residuals of $SO_2$ were found over clean areas and negative $SO_2$ SCDs are found at
high SZA which need to be corrected (note that this is a common problem of most
algorithms to retrieve $SO_2$ from space UV sensors). A background correction scheme was
found mostly necessary for the $SO_2$ slant columns retrieved in the baseline fitting window.
The adopted correction scheme depends on across-track position and measured $O_3$ slant
column as described below.
The correction is based on a parameterization of the background values that are then
subtracted from the measurements. The scheme first removes pixels with high SZA (>70°) or
SCDs larger than 1.5 DU (measurements with presumably real $SO_2$) and then calculates the
offset correction by averaging the $SO_2$ data on an ozone slant column grid (bins of 75 DU).
This is done independently for each across-track position and hemisphere, and the
correction makes use of measurements averaged over a time period of two weeks preceding
the measurement of interest (to improve the statistics and minimize the impact of a possible
extended volcanic $SO_2$ plume on the averaged values).
It should be noted that the $O_3$ slant column is dependent on the wavelength when applying
the approach of Puķīte et al. (2010):

$$SCD(\lambda) = SCD_{T1} + SCD_{T2} + \lambda.SCD_{\lambda} + \sigma_s(\lambda)SCD_S \qquad (4)$$



$SCD_{T1}$ and $SCD_{T2}$ are the retrieved ozone slant columns corresponding to the ozone cross-
sections at two temperatures (T1, T2) included in the fit. $SCD_{\lambda}$ and $SCD_s$ are the retrieved
parameters for the two pseudo cross-sections $\lambda.\sigma_s$ and $\sigma_s^2$ ($\sigma_s$ being the $O_3$ cross-section at
T1). In order to apply the background correction, the $O_3$ slant column expression (Eq. 4) is
evaluated at 313 nm (read below).
An example of the effect of the background correction is shown in Figure 4 for OMI. One can
see that after correction (top panel) the retrievals show smooth/unstriped results and values
close to zero outside the polluted areas. In some regions (in particular at high latitudes),
residual columns can be found, but are generally lower than 0.2 DU.
For the two additional fitting windows, residual $SO_2$ levels are relatively small in comparison
to the column amounts expected to be retrieved in these windows. However, simplified
background corrections are also applied to the alternative windows: the offset corrections
use parameterizations of the background slant columns based on latitude (bins of 5°), cross-
track position and time (two weeks moving averages as for the baseline window). To avoid
contamination by strong volcanic eruptions, only the pixels are kept with SCD less than 50DU
and 250DU for the fitting windows 325-335nm and 360-390nm, respectively.
It should be noted that the background corrections do not imply to save two weeks of $SO_2$ L2
data in intermediate products, but only the averaged values ($\Sigma_{i=1,N} SCD_i / N$) over the
predefined working grids (note: the numerators $\Sigma_{i=1,N} SCD_i$ and denominators N are stored
separately).
This background correction is well suited for the case of a 2D-detector array such as
TROPOMI, for which across-track striping can possibly arise due to imperfect cross-
calibration and different dead/hot pixel masks for the CCD detector regions. This
instrumental effect can also be found for scanning spectrometers, but since these
instruments only have one single detector, such errors do not appear as stripes. These
different retrieval artefacts can be compensated (up to a certain extent) using background
corrections which depend on the  across-track position. All of these corrections are also
meant to handle the time-dependent degradation of the instrument.  Note that experiences
with OMI show that the most efficient method to avoid across-track stripes in the retrievals
is to use row-dependent mean radiances as control spectrum in the DOAS fit.



### 2.2.3 Air mass factors
The DOAS method assumes that the retrieved slant column (after appropriate background
correction) can be converted into a vertical columns using a single air mass factor $M$
(representative for the fitting interval):

$$M = \frac{N_s}{N_v} \qquad (5)$$

which is determined by radiative transfer calculations with LIDORT version 3.3 (Spurr, 2008).
The AMF calculation is based on the formulation of Palmer et al. (2001):

$$M = \int m'(p) \cdot s(p)\,\mathrm{d}p \qquad (6)$$

with $m'=m(p)/C_{temp}(p)$, where m(p) is the so-called weighting function (WF) or pressure
dependent air mass factor, $C_{temp}$ is a temperature correction (see section 2.2.3.7) and $s$ is the
$SO_2$ normalized a-priori mixing ratio profile, as function of pressure ($p$).
The AMF calculation assumes Lambertian reflectors for the ground and the clouds and
makes use of pre-calculated WF LUTs at 313, 326 and 375 nm (depending on the fitting
window used). Calculating the AMF at these three wavelengths was found to give the best
results using closed-loop retrievals (see Auxiliary material of Theys et al., 2015). The WF
depends on observation geometry (solar zenith angle: SZA, line-of-sight angle: LOS, relative
azimuth angle: RAA), total ozone column (TO3), scene albedo (alb), surface pressure ($p_s$),
cloud top pressure ($p_{cloud}$) and effective cloud fraction ($f_c$).
Examples of $SO_2$ weighting functions are displayed in Figure 5 (as a function of height for
illustration purpose) and show the typical variations of the measurement sensitivity as a
function of height, wavelength and surface albedo.
The generation of the WF LUT has been done for a large range of physical parameters, listed
in Table 4. In practice, the WF for each pixel is computed by linear interpolation of the WF
LUT at the a-priori profile pressure grid and using the auxiliary data sets described in the
following sub-sections. Linear interpolations are performed along the cosine of solar and
viewing angles, relative azimuth angle and surface albedo, while a nearest neighbor
interpolation is performed in surface pressure. In particular, the grid of surface pressure is
very thin near the ground, in order to minimize interpolation errors caused by the generally



low albedo of ground surfaces. Furthermore, the LUT and model pressures are scaled to the
respective surface pressures, in order to avoid extrapolations outside the LUT range.
### 2.2.3.1 Observation geometry
The LUT covers the full range of values for solar zenith angles, line-of-sight angles and
relative azimuth angles that can be encountered in the TROPOMI measurements. The
observation geometry is readily present in the L1b data for each satellite pixel.
### 2.2.3.2 Total ozone column
The measurement sensitivity at 313 nm is dependent on the total ozone absorption. The LUT
covers a range of ozone column values from 200 to 500 DU for a set of typical ozone profiles.
The total ozone column is directly available from the operational processing of the S5P total
ozone column product.
### 2.2.3.3 Surface albedo
For the surface albedo dimension, we use the climatological monthly minimum Lambertian
equivalent reflector (minLER) data from Kleipool et al. (2008) at 328 nm for w1 and w2, and
376 m for w3. This database is based on OMI measurements and has a spatial resolution of
0.5° x 0.5°. The albedo value is very important for PBL anthropogenic $SO_2$ but less critical for
volcanic $SO_2$ when it is higher in the atmosphere.
### 2.2.3.4 Clouds
The AMF calculations for TROPOMI partly cloudy scenes use the cloud parameters (cloud
fraction, cloud albedo, cloud pressure) supplied by the nominal S5P cloud algorithm
OCRA/ROCINN in its Clouds as Reflecting Boundaries (CRB) implementation (Loyola et al.,
2016). The cloud surface is considered to be a Lambertian reflecting surface and the
treatment of clouds is achieved through the independent pixel approximation (IPA; Martin et
al., 2002) which considers a inhomogeneous satellite pixel as being composed (as for the
radiance intensity) of two independent homogeneous scenes, one completely clear and the
other completely cloudy. The weighting function is expressed as:

$$m(p) = \Phi m_{\text{cloud}}(p) + (1 - \Phi)m_{\text{clear}}(p) \tag{7}$$

where $\Phi$ is the intensity-weighted cloud fraction or cloud radiance fraction:



$$\Phi = \frac{f_c I_{cloud}}{f_c I_{cloud} + (1-f_c) I_{clear}}$$

(8)

The suffixes clear and cloudy refer to the WF and intensity calculation corresponding to a
fully clear or cloudy pixel, respectively. The WF LUT is therefore accompanied by an intensity
LUT with the same input grids. Both LUTs have been generated for a range of cloud cover
fractions and cloud top pressures.
Note that the variations of the cloud albedo are directly related to the cloud optical
thickness. Strictly speaking, in a Lambertian (reflective) cloud model approach, only thick
clouds can be represented. An effective cloud fraction corresponding to an effective cloud
albedo of 0.8 ($f_{eff} \cong f_c \frac{A_c}{0.8}$) can be defined, in order to transform optically thin clouds into
equivalent optically thick clouds of reduced extent. Note that in some cases (thick clouds
with $A_C > 0.8$) the effective cloud fraction can be larger than one and the algorithm assumes
$f_{eff}=1$. In such altitude dependent air mass factor calculations, a single cloud top pressure is
assumed within a given viewing scene. For low effective cloud fractions ($f_{eff}$ lower than
10%), the current cloud top pressure output is highly unstable and it is therefore reasonable
to consider the observation as a clear-sky pixel (i.e. the cloud fraction is set to 0 in Eq. 8) in
order to avoid unnecessary error propagation through the retrievals, which can be as high as
100%. Moreover, it has been shown recently by Wang et al. (2016) using multi-axis DOAS
(MAX-DOAS) observations to validate satellite data that in case of elevated aerosol loadings
in the PBL (typically leading to apparent $f_{eff}$ up to 10%), it is recommended to apply clear-
sky AMFs rather than total AMFs (based on cloud parameters) that presumably correct
implicitly for the aerosol effect on the measurement sensitivity.



It should be noted that the formulation of the pressure dependent air mass factor for a
partly cloudy pixel implicitly includes a correction for the $SO_2$ column lying below the cloud
and therefore not seen by the satellite, the so-called ghost column. Indeed, the total AMF
calculation as expressed by Eqs. 6 and 7 assumes the same shape factor and implies an
integration of the a-priori profile from the top of atmosphere to the ground, for each
fraction of the scene. The ghost column information is thus coming from the a-priori profile
shapes. For this reason, only observations with moderate cloud fractions ($f_{eff}$ lower than
30%) are used, unless it can be assumed that the cloud cover is mostly situated below the
$SO_2$ layer, i.e. a typical situation for volcanic plumes injected in the upper-troposphere or
lower-stratosphere.
2.2.3.5 Surface height
The surface height ($z_s$) is determined for each pixel by interpolating the values of a high
resolution digital elevation map, GMTED2010 (Danielson et al., 2011).
2.2.3.6 Profile shapes
It is generally not possible to know at the time of observation what is the $SO_2$ vertical profile
and whether the observed $SO_2$ is of volcanic origin or from pollution (or both). Therefore, the
algorithm computes four vertical columns for different hypothetical $SO_2$ profiles.
Three box profiles of 1 km thickness, located in the boundary layer, upper-troposphere and
lower-stratosphere, are used. The first box profile stands for typical conditions of well mixed
$SO_2$ (from volcanic or anthropogenic emissions) in the boundary layer while the upper-
troposphere and lower stratosphere box profiles are representative of volcanic $SO_2$ plumes
from effusive and explosive eruptions, respectively.
In order to have more realistic $SO_2$ profiles for polluted scenes, daily forecasts calculated
with the global TM5 chemical transport model (Huijnen et al., 2010) will be used. TM5 will
be operated with a spatial resolution of 1°x1° in latitude and longitude, and with 34 sigma
pressure levels up to 0.1 hPa in the vertical direction. TM5 will use 3-hourly meteorological
fields from the European Centre for Medium Range Weather Forecast (ECMWF) operational
model (ERA-Interim reanalysis data for reprocessing, and the operational archive for real
time applications and forecasts). These fields include global distributions of wind,
temperature, surface pressure, humidity, (liquid and ice) water content, and precipitation. A



more detailed description of the TM5 model is given at http://tm.knmi.nl/ and by van Geffen
et al. (2016).
For the calculation of the air mass factors, the profiles are linearly interpolated in space and
time, at the pixel centre and S5P local overpass time, through a model time step of 30
minutes. For NRT processing, the daily forecast of the TM5 model (located at KNMI) will be
ingested by the UPAS operational processor.
To reduce the errors associated to topography and the lower spatial resolution of the model
compared to the TROPOMI 7x3.5 km$^2$ spatial resolution, the a-priori profiles need to be
rescaled to effective surface elevation of the satellite pixel. The TM5 surface pressure is
converted by applying the hypsometric equation and the assumption that temperature
changes linearly with height (Zhou et al., 2009):

$$p_s = p_{TM5}\left(\frac{T_{TM5}}{(T_{TM5} + \Gamma(z_{TM5} - z_s))}\right)^{-\frac{g}{R\Gamma}} \tag{9}$$

where $p_{TM5}\ and\ T_{TM5}$ are the TM5 surface pressure and temperature, $\Gamma = 6.5 \text{Kkm}^{-1}$ the
lapse rate, $z_{TM5}$ the TM5 terrain height, and $z_s$ surface elevation for the satellite ground
pixel.
2.2.3.7 Temperature correction
The SO$_2$ absorption cross-sections of Bogumil et al. (2003) show a clear temperature
dependence which has an impact on the retrieved SO$_2$ SCDs depending on the fitting
window used. However, only one temperature (203K) is used for the DOAS fit, therefore a
temperature correction needs to be applied: SCD'=C$_{temp}$.SCD.  While the SO$_2$ algorithm
provides vertical column results for a set of a-priori profiles, applying this correction to the
slant column is not simple and as a workaround it is preferred to apply the correction
directly to the AMFs (or box-AMFs to be precise) while keeping the (retrieved) SCD
unchanged: AMF'=AMF/C$_{temp.}$ This formulation implicitly assumes that the AMF is not
strongly affected by temperature, which is a reasonable approximation (optically thin
atmosphere). The correction to be applied requires a temperature profile for each pixel
(which is obtained from the TM5 model):



$$C_{temp} = 1/[1 - \alpha.(T[K] - 203)] \qquad (10)$$
where $\alpha$ equals 0.002, 0.0038 and 0, for the fitting windows 312-326 nm, 325-335 nm and
360-390 nm, respectively. The parameter $\alpha$ has been determined empirically by fitting Eq. 10
through a set of data points (Figure 6), for each fitting window. Each value in Figure 6 is the
slope of the fitting line between the $SO_2$ differential cross-sections at 203K vs the cross-
section at a given temperature. In the fitting window 360-390 nm, no temperature
correction is applied ($\alpha$=0) because the cross-sections are quite uncertain. Moreover, the
360-390 nm wavelength range is meant for extreme cases (strong volcanic eruptions) for $SO_2$
plumes in the lower-stratosphere where a temperature of 203K is a good baseline.
### 2.2.3.8 Aerosols
The presence of aerosol in the observed scene (likely when observing anthropogenic
pollution or volcanic events), may affect the quality of the $SO_2$ retrieval (e.g. Yang et al.,
2010). No explicit treatment of aerosols (absorbing or not) is foreseen in the algorithm as
there is no general and easy way to treat the aerosols effect on the retrieval. At processing
time, the aerosol parameters (e.g., extinction profile or single scattering albedo) are
unknown. However, the information on the S5P UV Absorbing Aerosol Index (AAI) by Zweers
et al. (2016) will be included in the L2 $SO_2$ files as it gives information to the users on the
presence of aerosols both for anthropogenic and volcanic $SO_2$. Nevertheless, the AAI data
should be used/interpreted with care. In an offline future version of the $SO_2$ product,
absorbing aerosols might be included in the forward model, if reliable information on
absorbing aerosol can be obtained from the AAI and the S5P aerosol height product (Sanders
et al., 2016).





**3. ERROR ANALYSIS**
**3.1 INTRODUCTION**
The total uncertainty (accuracy and precision) on the $SO_2$ columns produced by the
algorithm presented in section 2, is composed of many sources of error (see also e.g., Lee et
al., 2009). Several of them are related to the instrument, such as uncertainties due to noise
or knowledge of the slit function. These instrumental errors propagate into the uncertainty
on the slant column. Other types of error can be considered as model errors and are related
to the representation of the physics in the algorithm. Examples of model errors are
uncertainties on the trace gas absorption cross-sections and the treatment of clouds. Model
errors can affect the slant column results or the air mass factors.
The total retrieval uncertainty on the $SO_2$ vertical columns can be derived by error
propagation, starting from Eq. 1 and if one assumes uncorrelated retrieval steps (Boersma et
al., 2004; De Smedt et al., 2008):

$$\sigma_{N_V}^2 = \left(\frac{\sigma_{N_S}}{M}\right)^2 + \left(\frac{\sigma_{N_S^{back}}}{M}\right)^2 + \left(\frac{(N_S - N_S^{back})\sigma_M}{M^2}\right)^2 \qquad (11)$$

where $\sigma_{N_S}$ and $\sigma_{N_S}^{back}$ are the errors on the slant column $N_S$ and on the background correction
$N_S^{back}$, respectively.
The error analysis is complemented by the total column averaging kernel (AK) as described in
Eskes and Boersma (2003):

$$AK(p) = \frac{m'(p)}{M} \qquad (12)$$

which is if often used to characterize the sensitivity of the retrieved column to a change in
the true profile.
**3.2 ERROR COMPONENTS**
The following sections describe and characterize 20 error contributions to the total $SO_2$
vertical column uncertainty. These different error components and corresponding typical
values are summarized in Tables 5 and 6. Note that, at the time of writing, the precise effect
of several S5P-specific error sources are unknown and will be estimated during operations.



A difficulty in the error formulation presented above comes from the fact that it assumes the
different error sources/steps of the algorithm to be independent and uncorrelated, which is
not strictly valid. For example, the background correction is designed to overcome
systematic features/deficiencies of the DOAS slant column fitting and these two steps
cannot be considered as independent. Hence, summing up all the corresponding error
estimates would lead to overestimated error bars. Therefore, several error sources will be
discussed in the following sub-sections without giving actual values at this point. Their
impact is included and described in later sub-sections.
Another important point to note is that one should also (be able to) discriminate systematic
and random components of a given error source V:

$$\sigma_V^2 = \frac{\sigma_{V(rand)}^2}{n} + \sigma_{V(syst)}^2 \tag{13}$$

here $n$ is the number of pixels considered. However, they are hard to separate in practice.
Therefore, each of the 20 error contributions are (tentatively) classified as either "random"
or "systematic" errors, depending on their tendencies to average out in space/time or not.
**3.2.1   Errors on the slant column**
Error sources that contribute to the total uncertainty on the slant column originate both
from instrument characteristics and uncertainties/limitations on the representation of the
physics in the DOAS slant column fitting algorithm. For the systematic errors on the slant
column, the numbers provided in Table 5 have been determined based on sensitivity tests
(using the QDOAS software).
All effects summed in quadrature, the various contributions are estimated to account for a
systematic error of about 20% +0.2DU of the background-corrected slant column ($\sigma_{N_s,syst} =$
$0.2 * (N_s - N_s^{back})$+0.2DU).
For the random component of the slant column errors, the error on the slant columns
provided by the DOAS fit is considered (hereafter referred to as SCDE) as it is assumed to be
dominated by and representative for the different random sources of error.
*Error source 1: SO$_2$ cross-section*



Systematic errors on slant columns due to $SO_2$ cross-sections uncertainties are estimated to
be around 6% (Vandaele et al., 2009) in window 1 (312-326 nm) and window 2 (325-335 nm)
and unknown in window 3 (360-390 nm). In addition, the effect of the temperature on the
$SO_2$ cross-sections has to be considered as well. We refer to see section 3.2.2 for a discussion
of this source of error.
*Error source 2: $O_3$ and $SO_2$ absorption*
Non-linear effects due to $O_3$ absorption are to a large extent accounted for using the Taylor
expansion of the $O_3$ optical depth (Pukīţe et al., 2010). Remaining systematic biases are then
removed using the background correction; hence residual systematic features are believed
to be small (please read also the discussion on errors 9 and 10). The random component of
the slant column error contributes to SCDE.
Non-linear effects due to $SO_2$ absorption itself (mostly for volcanic plumes) are largely
handled by the triple windows retrievals but - as will be discussed in section 4 - the transition
between the different fitting windows is a compromise and there are cases where saturation
can still lead to rather large uncertainties. However, those are difficult to assess on a pixel to
pixel basis.
*Error source 3: Other atmospheric absorption/interferences*
In some geographical regions, several systematic features in the slant columns remain after
the background correction procedure (see discussion on error 9: background correction
error) and are attributed to spectral interferences not fully accounted for in the DOAS
analysis, such as incomplete treatment of the Ring effect. This effect has also a random
component and contributes to the retrieved SCD error (SCDE).
*Error source 4 : Radiance shot noise*
It has a major contribution to the SCDE and it can be estimated from typical S/N values of
S5P in UV band 3 (800-1000, according to Veefkind et al., 2012). This translates to typical
SCD random errors of about 0.3-0.5, 5 and 60  DU for window 1, 2 and 3, respectively. Note
that real measurements are needed to consolidate these numbers.
*Error source 5 : DOAS settings*



Tests on the effect of changing the lower and upper limits of the fitting windows by 1 nm
and the order of the closure polynomial (4 instead of 5) have been performed. Based on a
selection of orbits for the Kasatochi eruption (wide range of measured SCDs), the
corresponding SCD errors are less than 11, 6 and 8 % for window 1, 2 and 3, respectively.
*Error source 6: Wavelength and radiometric calibration*
Tests on the effect of uncertainties in the wavelength calibration have been performed in
the ESA CAMELOT study. The numbers are for a shift of 1/20th of the spectral sampling in
the solar spectrum and 1/100th of the spectral sampling in the Earthshine spectrum. The
shift can be corrected for, but interpolation errors can still lead to a remaining uncertainty of
a few percent.
Regarding radiometric calibration, the retrieval result is in principle insensitive to flat
(spectrally constant) offsets on the measured radiance because the algorithm includes an
intensity offset correction. From the ESA ONTRAQ study it was found that additive error
signals should remain within 2% of the measured spectrum.
*Error source 7: Spectral response function*
Uncertainties in the S5P instrumental slit functions can lead to systematic errors on the
retrieved $SO_2$ slant columns (to be determined).
*Error source 8: Other spectral features*
When additional spectral features of unknown origin are present in the measured spectrum,
the impact on the retrieved slant column values can be considerable. In the ONTRAQ study,
testing sinusoidal perturbation signals showed that this effect on the retrieval result
depends strongly on the frequency of the signal. Additives signals with an amplitude of 0.05
% of the measurement affect the retrieved $SO_2$ slant column up to 30%. The effect scales
more or less linearly with the signal amplitude.
*Error source 9: Background/destriping correction*
This error source is mostly systematic and important for anthropogenic $SO_2$ or for
monitoring degassing volcanoes. Based on OMI and GOME-2 test retrievals, the uncertainty
on the background correction is estimated to be < 0.2 DU. This value accounts for limitations



of the background correction in some clean areas  (e.g. above the Sahara) where residual
slant columns values are typically found (after correction), or for a possible contamination by
volcanic $SO_2$, after a strong eruption.
### 3.2.2   Errors on the air mass factor
The error estimates on the AMF are listed in Table 6 and are based on simulations and
closed-loop tests using the radiative transfer code LIDORT. One can identify two sources of
errors on the AMF. First, the adopted LUT approach has limitations in reproducing the
radiative transfer in the atmosphere (forward model errors). Secondly, the error on the AMF
depends on input parameter uncertainties. This contribution can be broken down into a
squared sum of terms (Boersma et al., 2004):

$$\sigma_M^2 = \left(\frac{\partial M}{\partial \text{alb}} \cdot \sigma_{\text{alb}}\right)^2 + \left(\frac{\partial M}{\partial \text{ctp}} \cdot \sigma_{\text{ctp}}\right)^2 + \left(\frac{\partial M}{\partial fc} \cdot \sigma_{fc}\right)^2 + \left(\frac{\partial M}{\partial s} \cdot \sigma_{s}\right)^2 \tag{14}$$

where $\sigma_{\text{alb}}$, $\sigma_{\text{ctp}}$, $\sigma_f$, $\sigma_s$ are typical uncertainties on the albedo, cloud top pressure, cloud
fraction and profile shape, respectively.
The contribution of each parameter to the total air mass factor error depends on the
observation conditions. The air mass factor sensitivities ($\frac{\partial M}{\partial parameter}$), i.e. the air mass factor
derivatives with respect to the different input parameters, can be derived for any particular
condition of observation using the altitude-dependent AMF LUT, created with LIDORTv3.3,
and using the a-priori profile shapes. In practice, a LUT of AMF sensitivities has been created
using reduced grids from the AMF LUT and a parameterization of the profile shapes based on
the profile shape height.



*Error source 10: AMF wavelength dependence*
Because of strong atmospheric absorbers (mostly ozone) and scattering processes, the $SO_2$
AMF shows a wavelength dependence. We have conducted sensitivity tests to determine the
optimal wavelengths for AMF calculations representative for each of the three fitting
windows. To do so, synthetic radiances and $SO_2$ SCDs have been generated using LIDORT for
typical observations scenarios and at spectral resolution and sampling compatible with S5P.
The spectra have been analyzed by DOAS and the retrieved SCDs have been compared to the
calculated SCDs at different wavelengths. It comes out of this exercise that 313, 326 and 375
nm provide the best results, for window 1, 2 and 3, respectively. Figure 7 shows an
illustration of these sensitivity tests in the baseline window; an excellent correlation and
slope close to 1 is found for the scatter plot of retrieved versus simulated slant columns
using an effective wavelength of 313 nm for the AMF. Overall, for low solar zenith angles,
the deviations from the truth are less than 5% in most cases, except for boundary layer (BL)
$SO_2$ at a 1 DU column level and for low albedo scenes (deviations up to 20%). For high solar
zenith angles deviations are less than 10% in most cases, except for BL $SO_2$ at a 1 DU column
level and for low albedo scenes (underestimation up to a factor of 2).
*Error source 11: Model atmosphere*
This error relates to uncertainties in the atmospheric profiles used as input of LIDORT for the
weighting function look-up-table calculations.
Although the effect of $O_3$ absorption on the AMF is treated in the algorithm, the $O_3$ profiles
used as input of LIDORT are not fully representative of the real profiles and typical errors
(including error due to interpolation) of 5-10% can occur.
A test has been performed by replacing the US standard atmosphere pressure and
temperature profiles by high latitude winter profiles and the impact on the results is found
to be small.
*Error source 12 : Radiative transfer model*
It is believed to be small, less than 5% (Hendrick et al., 2006; Wagner et al., 2007).
*Error source 13 : Surface albedo*



A typical uncertainty on the albedo is 0.02 (Kleipool et al., 2008). This translates to an error
on the air mass factor after multiplication by the slope of the air mass factor as a function of
the albedo (Eq. 14) and can be evaluated for each satellite pixel. As an illustration, Figure 8
shows the expected dependence of the AMF with albedo and also with the cloud conditions.
From Figure 8a, one concludes that the retrievals of $SO_2$ in the BL are much more sensitive to
the exact albedo value than for $SO_2$ higher up in the atmosphere, for this particular example.
More substantial errors can be introduced if the real albedo differs considerably from what
is expected, for example in the case of the sudden snowfall or ice cover. The snow/ice cover
flag in the L2 file will therefore be useful for such cases.
*Error source 14: Cloud fraction*
An uncertainty on the cloud fraction of 0.05 is considered. The corresponding AMF error can
be estimated through Eq.14 (see Figure 8b) or by analytic derivation from Eqs. 6-8.
*Error source 15: Cloud top pressure*
An uncertainty on the cloud top height of 0.5 km (~50 hPa) is assumed. The corresponding
AMF error can be estimated through Eq. 14. Figure 8c illustrates the typical behaviour of
signal amplification /shielding for a cloud below/ above the $SO_2$ layer. One can see that the
error (slope) dramatically increases when the cloud is at a height similar to the $SO_2$ bulk
altitude.
*Error source 16 : Cloud correction*
Sensitivity tests showed that applying the independent pixel approximation or assuming
cloud-free pixels makes a difference of only 5% on yearly averaged data (for anthropogenic
BL $SO_2$ VC with cloud fractions less than 40%).
*Error source 17: Cloud model*
Cloud As Layer (CAL) is the baseline of the S5P cloud algorithm, but a Lambertian Equivalent
Reflector (LER) implementation will be used for $NO_2$, $SO_2$ and HCHO retrievals. The error due
to the choice of the cloud model will be evaluated during the operational phase.
*Error source 18: Profile shape*





A major source of systematic uncertainty for most $SO_2$ scenes is the shape of the vertical $SO_2$
distribution. The corresponding AMF error can be estimated through Eq. 14 and estimation
of uncertainty on the profile shape. Note that vertical columns are provided with their
averaging kernels, so that column data might be improved for particular locations by using
more accurate $SO_2$ profile shapes based on input from models or observations.
For anthropogenic $SO_2$ under clear-sky conditions, sensitivity tests using a box profile from 0
to 1±0.5 km above ground level, or using the different profiles from the CAMELOT study
(Levelt et al., 2009), give differences in AMFs in the range of 20-35%. Note that for particular
conditions $SO_2$ may also be uplifted above the top of the boundary layer and sometimes
reach upper-tropospheric levels (e.g., Clarisse et al., 2011). $SO_2$ weighting functions displayed
in Figure 5 show that the measurement sensitivity is then increased up to factor of 3 and
therefore constitutes a major source of error.
In the $SO_2$ algorithm, the uncertainty on the profile shape is estimated using one parameter
describing the shape of the TM5 profile: the profile height, i.e. the altitude (pressure) below
which resides 75% of the integrated $SO_2$ profile. $\frac{\partial M}{\partial s}$ is approached by $\frac{\partial M}{\partial s_h}$ where $s_h$ is half of
the profile height. Relatively small variations of this parameter have a strong impact on the
total air mass factors for low albedo scenes, because altitude-resolved air mass factors
decrease strongly in the lower troposphere, where the $SO_2$ profiles peak (see e.g. Figure 5).
For volcanic $SO_2$, the effect of the profile shape uncertainty depends on the surface or cloud
albedo. For low albedo scenes (Fig 5a), if no external information on the $SO_2$ plume height is
available, it is a major source of error at all wavelengths. Vertical columns may vary up to a
factor of 5. For high albedo scenes (Fig 5b), the error is less than 50%. It should be noted that
these conditions are often encountered for strong eruptions injecting $SO_2$ well above the
cloud deck (high reflectivity). Further uncertainty on the retrieved $SO_2$ column may arise if
the vertical distribution shows distinct layers at different altitudes, due to the different
nature of successive phases of the eruption.
In the $SO_2$ algorithm, three 1km thick box profiles are used in the AMF calculations, mostly to
represent typical volcanic $SO_2$ profiles. The error due to the profile shape uncertainty is
estimated by varying the box center levels by 100 hPa.





*Error source 19: Aerosols*
The effect of aerosols on the air mass factors are not explicitly considered in the $SO_2$
retrieval algorithm. To some extent, however, the effect of the non-absorbing part of the
aerosol extinction is implicitly included in the cloud correction (Boersma et al., 2004).
Indeed, in the presence of aerosols, the cloud detection algorithm is expected to
overestimate the cloud fraction, resulting partly in a compensation effect for cases where
aerosols and clouds are at similar heights. Absorbing aerosols have a different effect on the
air mass factors, and can lead to significant errors for high aerosol optical depths (AODs). In
the TROPOMI $SO_2$ product, the absorbing aerosol index field can be used to identify
observations with elevated absorbing aerosols.
Generally speaking, the effect of aerosols on AMF is highly variable and strongly depends on
aerosols properties (AOD, height and size distribution, single scattering albedo, scattering
phase function, etc.). Typical AMFs uncertainties due to aerosols found in the literature are
given in Table 6. As aerosols affect cloud fraction, cloud top height and to some extent the
albedo database used, correlations between uncertainties on these parameters are to be
expected.
*Error source 20: Temperature correction*
The DOAS scheme uses an $SO_2$ cross-section at only one temperature (Bogumil et al., 2003,
at 203K) which is in general not representative of the effective temperature corresponding
to the $SO_2$ vertical profile. This effect is in principle accounted for by the temperature
correction (which is applied in practice to the AMFs , see section 2.2.3.7) but with a certain
error associated of ~5%.
**4. VERIFICATION**
The $SO_2$ retrieval algorithm presented in section 2, and hereafter referred as 'prototype
algorithm', has been applied to OMI and GOME-2 spectra. The results have been extensively
verified and validated against different satellite and ground-based data sets (e.g., Theys et
al., 2015; Fioletov et al., 2016; Wang et al., 2016). Here we report on further scientific
verification activities that took place during the ESA S5P L2WG project.



In addition to the prototype algorithm, a scientific algorithm (referred as 'verification
algorithm') has been developed in parallel. Both algorithms have been applied to synthetic
and real (OMI) spectra and results were compared. In this study, we only present and discuss
a selection of results (for OMI).
**4.1 VERIFICATION ALGORITHM**
The S5P TROPOMI Verification Algorithm was developed in close cooperation between the
*Max Planck Institute for Chemistry* (MPIC) in Mainz (Germany) and the *Institut für Methodik*
*und Fernerkundung* as part of the *Deutsches Institut für Luft- und Raumfahrt*
*Oberpfaffenhofen* (DLR-IMF). Like the prototype algorithm (PA), the verification algorithm
(VA) uses a multiple fitting window DOAS approach to avoid non-linear effects during the
SCD retrieval in case of high $SO_2$ concentrations in volcanic plumes. However, especially the
alternatively used fitting windows differ strongly from the ones used for the PA and are
entirely located in the lower UV range:
• 312.1-324 nm (*standard retrieval - SR*): Similar to baseline PA fitting window, ideal for
small columns
• 318.6-335.1 nm (*medium retrieval - MR*): This fitting window is essentially located in
between the first and second fitting window of the PA and was mainly introduced to
guarantee a smoother transition between the baseline window and the one used for
high $SO_2$ concentrations. The differential $SO_2$ spectral features are still about one
order of magnitude smaller than in the baseline window.
• 323.1-335.1 nm (*alternative retrieval - AR*): Similar to the intermediate fitting window
of the PA. This fitting window is used in case of high $SO_2$ concentrations. Although it
is expected that volcanic events with extreme $SO_2$ absorption are still affected by
non-linear absorption in this window, the wavelength range is sufficient for most
volcanic events.



Furthermore, the VA selection criteria for the transition from one window to another are
not just based on fixed $SO_2$ SCD thresholds. The algorithm allows for a slow and smooth
transition between different fit ranges by linearly decreasing the weight of the former
fitting window and at the same time increasing the weight of the following fitting
window:
1)  for $SO_2$ SCD $\leq 4 \times 10^{17}$ molec/cm² ($\approx$ 15 DU):

$$SO_2\ SCD = SR$$

2)  for $4 \times 10^{17}$ molec/cm² < $SO_2$ SCD < $9 \times 10^{17}$ molec/cm²:

$$SO_2\ SCD = SR * \left[1 - \frac{SR}{9 \times 10^{17} molec/cm^2}\right] + MR * \left[\frac{SR}{9 \times 10^{17} molec/cm^2}\right]$$

3)  for $SO_2$ SCD $\geq 9 \times 10^{17}$ molec/cm² ($\approx$ 33 DU):

$$SO_2\ SCD = MR$$

4)  for $9 \times 10^{17}$ molec/cm² < $SO_2$ SCD < $4.6 \times 10^{18}$ molec/cm²:

$$SO_2\ SCD = MR * \left[1 - \frac{MR}{4.6 \times 10^{18} molec/cm^2}\right] + AR * \left[\frac{AR}{4.6 \times 10^{18} molec/cm^2}\right]$$

5)  for $SO_2$ SCD $\geq 4.6 \times 10^{18}$ molec/cm² ($\approx$171 DU):

$$SO_2\ SCD = AR$$

To convert the final $SO_2$ SCDs into vertical column densities, a single-wavelength AMF for
each of the three fitting windows ($SO_2$ SR, MR and AR) is calculated using the LIDORT LRRS
v2.3 (Spurr et al., 2008). The AMF depends on the viewing angles and illumination, surface
and cloud conditions as well as on the $O_3$ total column, which is taken from the $O_3$ total
column retrieval. A cloudy and clear-sky AMF is calculated using temperature dependent
cross-sections for $SO_2$ (Bogumil et al., 2003) and $O_3$ (Brion et al., 1983): $AMF(\lambda) = \dfrac{\ln\left(\frac{I + SO2}{I - SO2}\right)}{\tau_{SO2}}$





with ($I_{+SO2}$) and ($I_{-SO2}$) being simulated Earthshine spectra with and without including $SO_2$ as a
trace gas, respectively. Both AMFs are combined using the cloud fraction information. Like
the PA, the VA is calculated for different a-priori $SO_2$ profiles (centre of mass at 2.5 km, 6 km
and 15 km) and a temperature correction is applied (see Section 2.2.3.7). In contrast to the
PA the VA uses Gaussian-shaped $SO_2$ profiles with a FWHM of 2.5km rather than box profiles
as in the PA. This choice however has only a minor influence on the AMF.
For further details on the VA, the reader is referred to the S5P Science Verification Report
(available          at:          https://earth.esa.int/web/sentinel/user-guides/sentinel-5p-
tropomi/document-library/-/asset_publisher/w9Mnd6VPjXlc/content/sentinel-5p-tropomi-
science-verification-report) for more detailed description and results.
**4.2 VERIFICATION RESULTS**
For the inter-comparison, the prototype algorithm and verification algorithm were applied
to OMI data for three different $SO_2$ emission scenarios: moderate volcanic $SO_2$ VCDs on May
1, 2005, caused by the eruption of the Anatahan volcano, elevated anthropogenic SO2 VCDs,
on May 1, 2005, from the Norilsk copper smelter (Russia), and strongly enhanced $SO_2$ VCDs,
on August 8, 2008, after the massive eruption of Mount Kasatochi.
In the following, both algorithms use the same assumption of an $SO_2$ plume located at 15 km
altitude for the AMF calculation. Even if this choice is not realistic for some of the presented
scenarios, it minimizes the influence of differences in the a-priori settings. Main deviations
between Prototype and Verification Algorithm are therefore expected to be caused by the
usage of different fit windows (determining their sensitivity and fit error) and especially the
corresponding transition criteria.
Figure 9 shows the resulting maps of the $SO_2$ VCD for the VA (upper panels) and PA (lower
panels) for the three selected test cases. As can be seen, both algorithms result in similar $SO_2$
VCDs, however, a closer look reveals some differences, such as the maximum VCDs which
are not necessarily appearing at the same locations. For the Anatahan case for instance, the
maximum VCD is seen closer to the volcano at the eastern end of the plume for the PA,
while it appears to be further downwind for the VA. This effect can be explained by the
corresponding fit windows used for both algorithms which may result in deviating $SO_2$ VCDs,



especially for SO$_2$ scenarios where the best choice is difficult to assess. This is illustrated in
Figure 10 showing scatter plots of VA versus PA SO$_2$ VCDs for the three test cases (Anatahan,
Norilsk and Kasatochi) color-coded differently depending on the fitting window used for VA
(left) and PA (right), respectively. While the PA uses strictly separated results from the
individual fit windows, the VA allows a smooth transition whenever resulting SO$_2$ SCDs are
found to be located in between subsequent fit ranges.
For all three test cases, it appears that the PA is less affected by data scattering for low SO$_2$
or SO$_2$ free measurements than the VA. For the shortest UV fit windows, both algorithms
mainly agree but VA VCDs tend to be higher by 10-15% than the PA VCDs for the Anatahan
and Kasatochi measurements but interestingly not for the Norilsk case. For SO$_2$ VCDs around
7 DU the PA seem to be slightly affected by saturation effects in 312-326 nm window while
VA already makes use of a combined SR/MR SCD. For larger SO$_2$ VCDs (> 10 DU), data sets
from both algorithms show an increased scattering, essentially resulting from the more
intensive use of fitting windows at longer wavelengths (for which the SO$_2$ absorption is
weaker). While it is difficult to conclude which algorithm is closer to the actual SO$_2$ VCDs, the
combined fit windows of the VA probably are better suited (in some SO$_2$ column ranges) for
such scenarios as the SO$_2$ cross-section is generally stronger for lower wavelength (< 325
nm) when compared to the intermediate fit window of the PA.
For extremely high SO$_2$ loadings, i.e. for the Kasatochi plume on August 8, 2008, the DOAS
retrievals from PA and VA require all three fit windows to prevent systematic
underestimation of the resulting SO$_2$ SCDs due to non-linear absorption caused by very high
SO$_2$ concentrations within the volcanic plume. Figure 9 (right panel) shows that the SO$_2$
distribution is similar for both algorithms, including the location of the maximum SO$_2$ VCD.
From Figure 10 (lowest panel), it can be seen that the VA shows higher values for SO$_2$ VCDs
<100 DU, for all three fit windows. For very high SO$_2$ VCDs, it seems that the Verification
Algorithm is already slightly affected by an underestimation of the SO$_2$ VCD caused by non-
linear radiative transfer effects in the SO$_2$ AR fit window, while the PA retrievals in the 360-
390 nm fit range are insensitive to saturation effects. We note, however, that the Kasatochi
plume contained also significant amounts of volcanic ash and we cannot rule out a possible
retrieval effect of volcanic ash on the observed differences between PA and VA SO$_2$ results.
Finally we have also investigated other cases with extreme concentrations of SO$_2$, and



contrasting results were found compared to the Kasatochi case. E.g., on September 4, 2014,
PA retrieved up to 260 DU of $SO_2$ during the Icelandic Bardarbunga fissure eruption while VA
only found 150 DU (not shown). Compared to Kasatochi, we note that this specific scenario
is very different as for the plume height (the $SO_2$ plume was typically in the lowermost
troposphere ~ 3km a.s.l.) and it is likely to play a role in the discrepancy between PA and VA
results.
In summary, we found that the largest differences between prototype and verification
algorithms are due to the fitting window transitions and differences of measurement
sensitivity of the fitting windows used (all subject differently to non-linear effects).
Verification results have shown that the prototype algorithm produces reasonable results for
all the expected scenarios, from modest to extreme $SO_2$ columns, and are therefore
adequate for treating the TROPOMI data. In a future processor update, the method could
however be refined.
**5.   VALIDATION OF TROPOMI $SO_2$  PRODUCT**
In this section, we give a brief summary of possibilities (and limitations) to validate the
TROPOMI $SO_2$ product with independent measurements.
Generally speaking, the validation of a satellite $SO_2$ column product is a challenge for several
reasons, on top of which is the representativeness of the correlative data when compared to
the satellite retrievals. Another reason comes from the wide range of $SO_2$ columns in the
atmosphere that vary from about 1DU level for anthropogenic $SO_2$ and low level volcanic
degassing to 10-1000 DU for medium to extreme volcanic explosive eruptions.
The space-borne measurement of anthropogenic $SO_2$ is difficult because of the low column
amount and reduced measurement sensitivity close to the surface. The $SO_2$ signal is covered
by the competing $O_3$ absorption and the column accuracy is directly affected by the quality
of the background correction applied. Among the many parameters of the $SO_2$ retrieval
algorithm that affect the results, the $SO_2$ vertical profile shape is of utmost importance for
any comparison with correlative data. The $SO_2$ column product accuracy is also directly
impacted by the surface albedo used as input for the AMF calculation, the cloud



correction/filtering and aerosols. In principle, all these effects will have to be addressed in
future validation efforts.
The measurement of volcanic $SO_2$ is facilitated by $SO_2$ columns often larger than for
anthropogenic $SO_2$. However, the total $SO_2$ column is strongly dependent on the height of
the $SO_2$ plume which is highly variable and usually unknown. For most volcanoes, there is no
ground-based equipment to measure $SO_2$ during an appreciable eruption and even if it is the
case, the data are generally difficult to use for validation. For strong eruptions, volcanic
plumes are transported over long-distances and can be measured by ground–based and
aircraft devices but generally there is only a handful of datasets available and the number of
coincidences is rather small.
For both anthropogenic and volcanic $SO_2$ measurements, the vertical distribution of $SO_2$ is a
key parameter limiting the product accuracy. If reliable (external) information on the $SO_2$
profile (or profile shape) is available, it is recommended to recalculate the $SO_2$ vertical
columns by using this piece of information and the column averaging kernels that can be
found in the TROPOMI $SO_2$ L2 files.
**5.1 GROUND-BASED MEASUREMENTS**
When considering the application of ground-based instruments for the validation of satellite
$SO_2$ observations, several types of instruments are to be considered.



Brewer instruments have the advantage to operate as part of a network
(http://www.woudc.org), but the retrieved $SO_2$ columns are generally found inaccurate for
the validation of anthropogenic $SO_2$. Yet in some cases they might be used for coincidences
with volcanic clouds, typically for $SO_2$ VCDs larger than 5-10 DU.
Multi-axis DOAS (MAX-DOAS) or direct-sun DOAS measurements (e.g., from Pandora
instruments) can be used to validate satellite $SO_2$ columns from anthropogenic emissions
(e.g., Theys et al., 2015; Jin et al., 2016; Wang et al., 2016), but cautiousness must be exerted
in the interpretation of the results because realistic $SO_2$ profile shapes must be used by the
satellite retrieval scheme. While direct-sun DOAS retrievals are independent of the $SO_2$
profile shape, MAX-DOAS observations carry information on the $SO_2$ vertical distribution but
it is not obvious that the technique is directly applicable to the validation of satellite $SO_2$
retrievals, because the technique is not able to retrieve the full $SO_2$ profile. Another
important limitation comes from the fact that ground-based DOAS and satellite instruments
have very different fields of view and are therefore probing different air masses. This can
cause large discrepancy between ground-based and satellite measurements in case of strong
horizontal gradients of the $SO_2$ column field.
DOAS instruments scanning through volcanic plumes are now routinely measuring volcanic
$SO_2$ emissions, as part of the Network for Observation of Volcanic and Atmospheric Change
(NOVAC; Galle et al., 2010), for an increasing number of degassing volcanoes. Ongoing
research focusses on calculating $SO_2$ fluxes from those measurements and accounting for
non-trivial radiative transfer effects (e.g. light dilution, see Kern et al., 2009). NOVAC flux
data could be used for comparison with TROPOMI $SO_2$ data but it requires techniques to
convert satellite $SO_2$ vertical column into mass fluxes (see e.g., Theys et al., 2013, and
references therein, Beirle et al., 2014).  Similarly, fast-sampling UV cameras are becoming
increasingly used to measure and invert $SO_2$ fluxes and are also relevant to validate
TROPOMI $SO_2$ data over volcanoes or anthropogenic point sources (e.g., power plants). It
should be noted, however, that ground-based remote-sensing instruments operating nearby
$SO_2$ point sources are sensitive to newly emitted $SO_2$ plumes while a satellite sensor like
TROPOMI will measure aged plumes that have been significantly depleted in $SO_2$. While in
some cases it is possible to compensate for this effect by estimating the $SO_2$ lifetime e.g.
directly from the space measurements (Beirle et al., 2014), the general situation is that the



$SO_2$ loss rate is highly variable (especially in volcanic environments) and this can lead to
strong discrepancies when comparing satellite and ground-based $SO_2$ fluxes.
In addition to optical devices, there are also in-situ instruments measuring surface $SO_2$
mixing ratios. This type of instrument can only validate surface concentrations, and
additional information on the $SO_2$ vertical profile (e.g., from model data) is required to make
the link with the satellite retrieved column. However, in-situ instruments are being operated
for pollution monitoring in populated areas, and allow for extended and long term
comparisons with satellite data (see e.g. Nowlan et al., 2011).
**5.2 AIRCRAFT AND MOBILE MEASUREMENTS**
Airborne and mobile instruments provide valuable and complementary data for satellite
validation.
In case of volcanic explosive eruptions, satisfactory validation results can be obtained by
comparing satellite and fixed ground DOAS measurements of drifting $SO_2$ plumes, as shown
by Spinei et al. (2008), but the comparison generally suffers from the small number of
coincidences. Dedicated aircraft campaign flights (e.g. Schumann et al., 2011) can in
principle improve the situation. Their trajectory can be planned with relative ease to cross
sustained eruptive plumes. However, localized high $SO_2$ concentrations, may be carried away
too quickly to be captured by aircraft or have diluted below the threshold limit for satellite
detection before an aircraft can respond. An important data base of $SO_2$ aircraft
measurements is provided by the CARIBIC/IAGOS project which exploits automated scientific
instruments operating long distance commercial flights. Measurements of volcanic $SO_2$
during the eruptions of Mt. Kasatochi and Eyjafjallajökull and comparison with satellite data
have been reported by Heue et al. (2010, 2011).





An attempt to validate satellite $SO_2$ measurements using mobile DOAS instrument for a fast
moving (stratospheric) volcanic $SO_2$ plume was presented by Carn and Lopez (2011).
Although the agreement between both data sets was found reasonable, the comparison was
complicated by the relatively fast displacement of the volcanic cloud with respect to the
ground spectrometer and clear heterogeneity on scales smaller than a satellite pixel. For
degassing volcanoes or newly fissure eruptions, mobile DOAS traverse measurements under
the plume offer unique opportunities to derive volcanic $SO_2$ fluxes that could be used to
validate satellite measurements.
For polluted regions, measurements of anthropogenic $SO_2$ by airborne nadir-looking DOAS
sensors are able to produce high spatial resolution mapping of the $SO_2$ column field (e.g.,
during the AROMAT campaigns, http://uv-vis.aeronomie.be/aromat/) that could be used to
validate TROPOMI $SO_2$ product or give information on horizontal gradients of the $SO_2$ field
(e.g. in combination with coincident mobile DOAS measurements) that would be particularly
useful when comparing satellite and MAX-DOAS data (see discussion in section 5.1). Equally
important are also limb-DOAS or in-situ instruments to provide information on vertical
distribution of $SO_2$ which is crucial for satellite validation (e.g., Krotkov et al., 2008).
**5.3 SATELLITE  MEASUREMENTS**
Inter-comparison of satellite $SO_2$ measurements generally provides a convenient and easy
way to evaluate at a glance the quality of a satellite product, by comparing $SO_2$ maps for
instance. Often, it also provides improved statistics and geographical representativeness but
it poses a number of problems because when different satellite sensors are compared they
have also different overpass times, swaths, spatial resolutions and measurement sensitivities
to $SO_2$.





For volcanic SO$_2$, satellite measurements often provide the only data available for the first
hours to days after an eruption event and satellite inter-comparison is thus the only practical
way to assess the quality of the retrievals. To overcome sampling issues mentioned above,
inter-comparison of SO$_2$ masses integrated over the measured volcanic plume is often
performed. For TROPOMI, current satellite instruments will be an important source of data
for cross-comparisons. Although non-exhaustive, the list of satellite sensors that could be
used is: OMI, OMPS, GOME-2 and IASI (MetOp-A, -B, and the forthcoming -C), AIRS, CrIS,
VIIRS and MODIS. As mentioned above, the inter-comparison of satellite SO$_2$ products is
difficult and in this respect the plume altitude is a key-factor of the satellite SO$_2$ data
accuracy. Comparison of TROPOMI and other satellite SO$_2$ products will benefit from the
advent of scientific algorithms for the retrieval of SO$_2$ plume heights but also from the use of
volcanic plume height observations using space lidar instruments (e.g. CALIOP and the future
EarthCare mission).
For both anthropogenic SO$_2$ and volcanic degassing SO$_2$, the satellite UV sensors OMI, GOME-
2 and OMPS can be compared to TROPOMI SO$_2$ data by averaging data over certain polluted
regions. It will give valuable information on the data quality but, in some cases, the
comparison will suffer from differences in spatial resolution. A more robust and in-depth
comparison would be to use different TROPOMI SO$_2$ datasets generated by different
retrieval algorithms and investigate the differences in the various retrieval steps (spectral
fitting, corrections, radiative transfer simulations, error analysis).

## 6  CONCLUSIONS

Based on the heritage from GOME, SCIAMACHY, GOME-2 and OMI, a DOAS retrieval
algorithm has been developed for the operational retrieval of SO$_2$ vertical columns from
TROPOMI Level1b measurements in the UV spectral range. Here we describe its main
features.
In addition to the traditionally used fitting window of 312-326 nm, the new algorithm allows
for the selection of two additional fitting windows (325-335 nm and 360-390nm), reducing
the risk of saturation and ensuring accurate SO$_2$ column retrieval even for extreme SO$_2$



concentrations as observed for major volcanic events. The spectral fitting procedure also
includes an advanced wavelength calibration scheme and a spectral spike removal algorithm.
After the slant column retrieval, the next step is a background correction, which is
empirically based on the $O_3$ slant column (for the baseline fitting window) and across-track
position, and accounts for possible across-track dependencies and instrumental degradation.
The $SO_2$ slant columns are then converted into vertical columns by the means of air mass
factor calculations. The latter is based on weighting function look-up-tables with
dependencies on the viewing geometry, clouds, surface pressure, albedo, ozone, and is
applied to pre-defined box profiles and TM5 CTM forecast profiles. In addition, the algorithm
computes DOAS-type averaging kernels and a full error analysis of the retrieved columns.
In this paper we have also presented verification results using an independent algorithm for
selected OMI scenes with enhanced $SO_2$ columns. Overall the prototype algorithm agrees
well with the verification algorithm, demonstrating its ability in retrieving accurately medium
to very high $SO_2$ columns. We have discussed the advantages and limitations of both
prototype and verification algorithms.
Based on the experience with GOME-2 and OMI, the TROPOMI $SO_2$ algorithm is expected to
have a comparable level of accuracy. Due to its high signal-to-noise ratio, TROPOMI will be
capable of at least achieving comparable retrieval precision as its predecessors but at a
much finer spatial resolution of 7x3.5 km² at best. For single measurements, the user
requirements for tropospheric $SO_2$ concentrations will not be met, but improved monitoring
of strong pollution and volcanic events will be possible by spatial and temporal averaging the
increased number of observations of TROPOMI. Nevertheless, it will require significant
validation work and here we have discussed some of the inherent challenges for both
volcanic and anthropogenic $SO_2$ retrievals. Correlative measurements from ground-based,
aircraft/mobile, and satellite instruments, will be needed over different regions and various
emission scenarios to assess and characterize the quality of TROPOMI $SO_2$ retrievals.





The baseline algorithm presented here, including all its modules (slant column retrieval,
background correction, air mass factor calculation and error analysis), has been fully
implemented in the S5P operational processor UPAS by the DLR team. Figure 11 illustrates
the status of the implementation for one day of OMI test data, exemplarily for the slant
columns retrievals. A nearly perfect agreement is found between SCD results over 4 orders
of magnitude. A similar match between prototype algorithm and operational processor is
found for all other retrieval modules.
For more information on the TROPOMI $SO_2$ L2 data files, the reader is referred to the S5P
$SO_2$ Product User Manual (Pedergnana et al., 2016).
**APPENDIX A. FEASIBILITY, INFORMATION ON DATA PRODUCT AND ANCILLARY DATA**
**High level data product description**
In addition to the main product results, such as $SO_2$ slant column, vertical column and air
mass factor, the level 2 data files will contain several additional parameters and diagnostic
information. Table A1 gives a minimum set of data fields that will be present in the Level 2
data. A 1-orbit $SO_2$ column Level 2 file will be of about 640 MB. More details about the
operational level 2 product based on the netCDF data format and the CF metadata
convention are provided in the $SO_2$ Product User Model (Pedergnana et al., 2016).
It should be noted that the averaging kernels are given only for the a-priori profiles from the
TM5 CTM (to save space). The averaging kernels for the box profiles can be estimated by
scaling the provided averaging kernel (corresponding to TM5 profiles): $AK_{box}(p)$
$=AK(p).Scaling box$. Following the AK formulation of Eskes and Boersma (2004), the scaling
factor is given simply by AMFs ratios: $AMF_{TM5}/AMF_{box}$.
**Auxiliary information**
The algorithm relies on several external data sets. These can be either static or dynamic. An
overview is given in Table A2 and A3.
**ACKNOWLEDGEMENTS**
This work has been performed in the frame of the TROPOMI project. We acknowledge
financial support from ESA S5P, Belgium Prodex TRACE-S5P projects, and Bayerisches





Staatsministerium fürWirtschaft und Medien, Energie und Technologie (grant 07 03/893 73/

2 5 /2013).

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



1  Table 1. Requirements on SO$_2$ vertical column products as derived from the MRTD. Numbers
2  denote accuracy / precision, respectively.

| | Horizontal resolution [km] | Required uncertainty | Achievable uncertainty | Theme (Table in MRTD) |
|---|---|---|---|---|
| Enhanced stratospheric column | 50-200 | 30% for VCD>0.5 DU | Met for VCD > 0.5DU | A3 |
| Tropospheric column | 5-20 | 30-60% or 1.3 x 10$^{15}$ molecules cm$^{-2}$ (least stringent) | 50% / 3-6 x 10$^{16}$ molec. cm$^{-2}$ | B1, B2, B3 |
| Total column | 5-20 | 30-60% or 1.3 x 10$^{15}$ molecules cm$^{-2}$ (least stringent) | 50% / 3-6 x 10$^{16}$ molec. cm$^{-2}$ | B1, B2, B3 |



1    Table 2. DOAS settings used to retrieved SO$_2$ slant columns

| | |
|---|---|
| ***Fitting intervals 1 and 2*** | 312-326 nm (w1), 325-335 nm (w2) |
| *Cross-sections* | SO$_2$: 203K (*Bogumil et al.*, 2003) |
| | O$_3$: 228K and 243K with *Io* correction (*Brion et al.*, 1998) |
| | Pseudo O$_3$ cross sections (λσ$_{O3}$, σ$_{O3}{}^2$) (*Puķīte et al.*, 2010) |
| | Ring effect: 2 eigenvectors (*Vountas et al.*, 1998) generated |
| | for 20° and 87° solar zenith angles using LIDORT-RRS (*Spurr et al.*, 2008) |
| *Polynomial* | 5$^{th}$ order |
| ***Fitting interval 3*** | 360-390 nm (w3) |
| *Cross-sections* | SO$_2$: *Hermans et al.* (2009) extrapolated at 203K |
| | NO$_2$: 220K (*Vandaele et al.*, 1998) |
| | O$_2$-O$_2$: *Greenblatt et al.*, 1990 |
| | Ring effect: single spectrum (*Chance and Spurr*, 1997) |
| *Polynomial* | 4$^{th}$ order |
| ***Intensity offset correction*** | Linear offset |
| ***Spectrum shift and stretch*** | Fitted |
| ***Spectral spikes removal procedure*** | *Richter et al.* [2011] |
| ***Reference spectrum*** | Baseline: Daily solar irradiance |
| | Foreseen update: Daily averaged earthshine spectrum in Pacific region (10°S-10°N, 160°E-120°W); separate spectrum for each detector row. NRT: averaged spectra of the last available day, Off-line: averaged spectra of the current day |





Table 3. Criteria for selecting alternative fitting windows.

| Window number | w1 | w2 | w3 |
|---|---|---|---|
| Wavelength range | 312 – 326 nm | 325-335 nm | 360-390 nm |
| Derived slant column | S1 | S2 | S3 |
| Application | Baseline for every pixel | S1 > 15 DU and S2 > S1 | S2 > 250 DU and S3 > S2 |



Table 4. Physical parameters that define the WF look-up table.

| Parameter | Number of grid points | Grid values | Symbol |
|---|---|---|---|
| Atmospheric pressure [hPa] | 64 | 1056.77, 1044.17, 1031.72, 1019.41, 1007.26, 995.25, 983.38, 971.66, 960.07, 948.62, 937.31, 926.14, 915.09, 904.18, 887.87, 866.35, 845.39, 824.87, 804.88, 785.15, 765.68, 746.70, 728.18, 710.12, 692.31, 674.73, 657.60, 640.90, 624.63, 608.58, 592.75, 577.34, 562.32, 547.70, 522.83, 488.67, 456.36, 425.80, 396.93, 369.66, 343.94, 319.68, 296.84, 275.34, 245.99, 210.49, 179.89, 153.74, 131.40, 104.80, 76.59, 55.98, 40.98, 30.08, 18.73, 8.86, 4.31, 2.18, 1.14, 0.51, 0.14, 0.03, 0.01, 0.001 | $p_l$ |
| Altitude corresponding to the atmospheric pressure, using an US standard atmosphere [km] | 64 | -0.35, -0.25, -0.15, -0.05, 0.05, 0.15, 0.25, 0.35, 0.45, 0.55, 0.65, 0.75, 0.85, 0.95, 1.10, 1.30, 1.50, 1.70, 1.90, 2.10, 2.30, 2.50, 2.70, 2.90, 3.10, 3.30, 3.50, 3.70, 3.90, 4.10, 4.30, 4.50, 4.70, 4.90, 5.25, 5.75, 6.25, 6.75, 7.25, 7.75, 8.25, 8.75, 9.25, 9.75, 10.50, 11.50, 12.50, 13.50, 14.50, 16.00, 18.00, 20.00, 22.00, 24.00, 27.50, 32.50, 37.50, 42.50, 47.50, 55.00, 65.00, 75.00, 85.00, 95.00 | $z_l$ |
| Solar zenith angle [°] | 17 | 0, 10, 20, 30, 40, 45, 50, 55, 60, 65, 70, 72, 74, 76, 78, 80, 85 | $\theta_0$ |
| Line of sight angle [°] | 10 | 0, 10, 20, 30, 40, 50, 60, 65, 70, 75 | $\theta$ |
| Relative azimuth angle [°] | 5 | 0, 45, 90, 135, 180 | $\phi$ |
| Total ozone column [DU] | 4 | 205, 295, 385, 505 | TO3 |
| Surface albedo | 14 | 0, 0.01, 0.025, 0.05, 0.075, 0.1, 0.15, 0.2, 0.25, 0.3 0.4, 0.6, 0.8, 1.0 | $A_s$ |
| Surface / cloud top pressure [hPa] | 17 | 1063.10, 1037.90, 1013.30, 989.28, 965.83, 920.58, 876.98, 834.99, 795.01, 701.21, 616.60, 540.48, 411.05, 308.00, 226.99, 165.79, 121.11 | $p_s$ |
| AMF Wavelength | 3 | 313, 326, 375 | |





Table 5. Systematic and random error components contributing to the total uncertainty on the $SO_2$ slant column.

| # | Error source | Type* | Parameter uncertainty | Typical uncertainty on $SO_2$ SCD |
|---|---|---|---|---|
| 1 | $SO_2$ absorption cross section | S | 6% (window 1)<br>6% (window 2)<br>unknown (window 3) | 6% |
| 2 | $SO_2$ and $O_3$ absorption | S & R | | Errors 9 & 10 |
| 3 | Other atmospheric absorption or interference | S & R | | Error 9 |
| 4 | Radiance shot noise | R | S/N=800-1000 | 0.3-0.5 DU (window 1)<br>5 DU (window 2)<br>60 DU (window 3) |
| 5 | DOAS settings | S | 1 nm, polynomial order | <11% (window 1)<br><6% (window 2)<br><8% (window 3) |
| 6 | Wavelength and radiometric calibration | S | Wavelength Calibration.<br><br>Radiometric calibration. Additive errors should remain below 2 %. | Wavelength calibration and spectral shifts can be corrected by the algorithm to less than 5 % effect on the slant column.<br><br>Intensity offset correction in principle treats (small) radiometric calibration errors |
| 7 | Spectral response function | | TBD | TROPOMI-specific<br>Expected uncertainty: 10% |
| 8 | Other spectral features | | Strongly dependent on interfering signal | - |
| 9 | Background correction | S & R | | 0.2 DU |

* R: random, S: systematic




Table 6. Systematic and random error components contributing to the total uncertainty on the $SO_2$ air mass factor.

| # | Error | Type* | Parameter uncertainty | Typical uncertainty on the AMF |
|---|-------|-------|----------------------|-------------------------------|
| 10 | AMF wavelength dependence | S | | 10% |
| 11 | Model atmosphere | S | $O_3$ profile<br>P,T profiles | ~5-10%<br>small |
| 12 | Forward model | S | < 5% | <5% |
| 13 | Surface albedo[†] | S | 0.02 | 15% (PBL)<br>5% (FT)<br>1% (LS) |
| 14 | Cloud fraction[†] | R | 0.05 | 5% (PBL)<br>15% (FT)<br>1% (LS) |
| 15 | Cloud top pressure[†] | R | 50 hPa | 50% (PBL)<br>50% (FT)<br>1% (LS) |
| 16 | Cloud correction | R | | < 5% on yearly averaged data |
| 17 | Cloud model | | TBD | |
| 18 | $SO_2$ profile shape | S | | anthropogenic $SO_2$<br>20%-35%<br><br>volcanic $SO_2$<br>large (low albedo), < 50% (high albedo) |
| 19 | Aerosol | S & R | | Anthropogenic $SO_2$ < 15% (Nowlan et al., 2011).<br>Volcanic $SO_2$ (aerosols: ash/sulphate) :<br>~ 20% (Yang et al., 2010) |
| 20 | Temperature correction | R | | ~5% |

* R: random, S: systematic        [†] Effect on the AMF estimated from Figure 6





1    Table A1. List of output fields in the TROPOMI SO$_2$ products. nAlong x nAcross corresponds

2    to the number of pixels in an orbit along track and across track, respectively.

| Name/Data | Symbol | Unit | Description | Data type | Number of entries per observation |
|---|---|---|---|---|---|
| **Date** | | n.u. | Date and time of the measurement YYMMDDHHMMSS.MS | characters | nAlong |
| **Latitudes** | $lat$ | degree | Latitudes of the four pixel corners + center | float | 5 x nAlong x nAcross |
| **Longitudes** | $lon$ | degree | Longitudes of the four pixel corners + center | float | 5 x nAlong x nAcross |
| **SZA** | $\theta_0$ | degree | Solar zenith angle | float | nAlong x nAcross |
| **VZA** | $\theta$ | degree | Viewing zenith angle | float | nAlong x nAcross |
| **RAA** | $\varphi$ | degree | Relative azimuth angle | float | nAlong x nAcross |
| **SCD** | $N_s$ | mol.m$^{-2}$ | SO2 slant column density | float | nAlong x nAcross |
| **SCDcorr** | $N_s^c$ | mol.m$^{-2}$ | SO2 slant column density background corrected | float | nAlong x nAcross |
| **VCD** | $N_v$ | mol.m$^{-2}$ | SO2 vertical column density (4values) | float | 4 x nAlong x nAcross |
| **Wdow flag** | $Wflag$ | n.u. | Flag for the fitting window used (1,2,3) | integer | nAlong x nAcross |
| **AMF** | $M$ | n.u. | Air mass factor (4values) | float | 4 x nAlong x nAcross |
| **Cloud free AMF** | $M_{clear}$ | n.u. | Cloud Free Air mass factor (4values) | float | 4 x nAlong x nAcross |
| **Cloudy AMF** | $M_{cloud}$ | n.u. | Fully Cloudy Air mass factor (4values) | float | 4 x nAlong x nAcross |
| **CF** | $f_c$ | n.u. | Cloud fraction | float | nAlong x nAcross |
| **CRF** | $\Phi$ | n.u. | Cloud radiance fraction | float | nAlong x nAcross |



| | | | | | |
|---|---|---|---|---|---|
| **CP** | $p_{cloud}$ | Pa | Cloud top pressure | float | nAlong x nAcross |
| **CH** | $z_{cloud}$ | m | Cloud top height | float | nAlong x nAcross |
| **CA** | $A_{cloud}$ | n.u. | Cloud top albedo | float | nAlong x nAcross |
| **Albedo** | $A_s$ | n.u. | Surface albedo | float | nAlong x nAcross |
| **Aerosol index** | AAI | n.u. | Absorbing Aerosol Index | float | nAlong x nAcross |
| **Ch-squared** | $Chi^2$ | n.u. | Chi-squared of the fit | float | nAlong x nAcross |
| **VCD error** | $\sigma\_N_v$ | mol.m$^{-2}$ | Total error on the vertical column (individual measurement) | float | 4x nAlong x nAcross |
| **SCD random error** | $\sigma\_N_{s\_rand}$ | mol.m$^{-2}$ | Random error on the slant column | float | nAlong x nAcross |
| **SCD systematic error** | $\sigma\_N_{s\_syst}$ | mol.m$^{-2}$ | Systematic error on the slant column | float | nAlong x nAcross |
| **AMF random error** | $\sigma\_M_{\_rand}$ | n.u. | Random error on the air mass factor (4values) | float | 4x nAlong x nAcross |
| **AMF systematic error** | $\sigma\_M_{\_syst}$ | n.u. | Systematic error on the air mass factor (4 values) | float | 4x nAlong x nAcross |
| **Averaging kernel** | AK | n.u. | Total column averaging kernel (for a-priori profile from CTM) | float | 34 x nAlong x nAcross |
| **Averaging kernel scalings for box profiles** | Scaling box | n.u. | Factors to apply to the averaging kernel function to obtain the corresponding averaging kernels for the 3 box profiles | float | 3x nAlong x nAcross |
| **SO$_2$ profile** | $n_a$ | n.u. | A-priori profile from CTM (volume mixing ratio) | float | 34 x nAlong x nAcross |
| **Surface altitude** | $z_s$ | m | Digital elevation map | float | nAlong x nAcross |
| **Surface pressure** | $p_s$ | Pa | Effective surface pressure of the satellite pixel | float | nAlong x nAcross |
| **TM5 level coefficient a** | $A_i$ | Pa | TM5 pressure level coefficients that effectively define the mid-layer levels | float | 24 |





| TM5 level coefficient b | $A_i$ | n.u. | (from ECMWF) | float | 24 |
|---|---|---|---|---|---|


1    Table A2. Static auxiliary data for the S5P SO$_2$ algorithm.

| Name/Data | Symbol | Unit | Source | Pre-process needs | Comments |
|---|---|---|---|---|---|
| **Absorption cross-sections** | | | | | |
| **SO2** | $\sigma_{SO2}$ | cm$^2$molec.$^{-1}$ | Bogumil et al. (2003), 203K, 223K, 243K, 293K  Hermans et al. (2009), all temperatures | Convolution at the instrumental spectral resolution using the provided slit function | |
| **Ozone** | $\sigma_{o3218}$  $\sigma_{o3243}$ | cm$^2$molec.$^{-1}$ | Brion et al. (1998) ; 218K and 243K. | | |
| **BrO** | $\sigma_{BrO}$ | cm$^2$molec.$^{-1}$ | Fleischmann et al. (2004), 223K | | |
| **NO$_2$** | $\sigma_{NO2}$ | cm$^2$molec.$^{-1}$ | Vandaele et al. (1998), 220K | | - |
| **O$_4$ (O$_2$-O$_2$)** | $\sigma_{O4}$ | cm$^5$molec.$^{-2}$ | Greenblatt et al. (1990) | | |
| **High resolution reference solar spectrum** | $E_s$ | W m$^{-2}$nm$^{-1}$ | Chance and Kurucz, 2010 | - | - |
| **Ring effect** | $\sigma_{ringev1}$  $\sigma_{ringev2}$ | cm$^2$molec.$^{-1}$ | 2 Ring cross-sections generated internally. | A high-resolution reference solar spectrum and the instrument slit function are needed to generate the data set. | Calculated in an ozone containing atmosphere for low and high SZA, using LIDORT_RRS (Spurr et al., 2008) and a standard atmosphere (Camelot European Pollution atmospheric profile). |
| **Non-linear O$_3$ absorption effect** | $\sigma_{o3l}$  $\sigma_{o3sq}$ | nm.cm$^2$molec.$^{-1}$  cm$^4$molec.$^{-2}$ | 2 pseudo-cross sections generated internally. | The O$_3$ cross-section at 218 K is needed. | Calculated from the Taylor expansion of the wavelength and the O$_3$ optical depth (Puķīte et al., 2010). |
| **Instrument slit function** | $SF$ | n.u. | Slit Function by wavelength/detector. | - | Values between 300 and 400nm. |
| **Surface Albedo** | $A_s$ | n.u. | OMI-based monthly minimum LER (update of Kleipool et al., 2008) | - | |
| **Digital elevation map** | $z_s$ | m | GMTED2010 (Danielson et al., 2011) | | Average over the ground pixel area. |
| **SO2 profile** | $n_a$ | n.u. | One kilometre thick box profiles, with three different peak altitudes, | - | TM5 profiles from the last available day in case theTM5 profiles of the |





| | | | | | |
|---|---|---|---|---|---|
| | | | representing different altitude regimes: Boundary layer: from the surface altitude to 1km above it. Free troposphere: centred around 7 km altitude. Lower stratosphere: centred around 15 km altitude. Daily $SO_2$ profiles forecast from TM5 | | current day are not available |
| Look-up table of pressure-resolved AMFs | $m$ | n.u. | Calculated internally with the LIDORTv3.3 RTM (Spurr, 2008). | - | For the different fitting windows (312-326 nm, 325-335 nm, 360-390 nm), the assumed vertical column is 5 DU, 100 DU, 500 DU, respectively. |
| Temperature correction parameters | α | $K^{-1}$ | Bogumil et al. (2003) | - | - |



1    Table A3. Dynamic auxiliary data for the S5P SO$_2$ algorithm.

| Name/Data | Symbol | Unit | Source | Pre-process needs | Backup if not available |
|---|---|---|---|---|---|
| S5P level 1B Earth radiance | $I$ | mol s$^{-1}$ m$^{-2}$nm$^{-1}$sr$^{-1}$ | S5P L1b product | - | No retrieval |
| S5P level 1B sun irradiance | $E_0$ | mol s$^{-1}$ m$^{-2}$nm$^{-1}$ | S5P L1b product | Wavelength recalibrated using a high-resolution reference solar spectrum | Use previous measurement |
| S5P Cloud fraction | $f_c$ | n.u. | S5 P operational cloud product based on a Lambertian cloud model (Loyola et al., 2016) UPAS processor. | - | No retrieval |
| S5P Cloud top pressure | $p_{cloud}$ | Pa | | | |
| S5P Cloud top albedo | $A_{cloud}$ | n.u. | | | |
| SO2 profile | $n_a$ | n.u. | Daily forecast from TM5 CTM run at KNMI. | - | Use TM5 CTM profile from last available day |
| Temperature profile | T | K | Daily forecast from TM5 CTM run at KNMI. | - | Use TM5 CTM profile from last available day |
| S5P Absorbing aerosol index | $AAI$ | n.u. | S5P operational AAI product (Zweers et al., 2016). Used for flagging. KNMI processor. | - | Missing information flag. |
| Snow-ice flag | | n.u. | Near real-time global Ice and Snow Extent (NISE) data from NASA. | - | Use snow/ice climatology. |





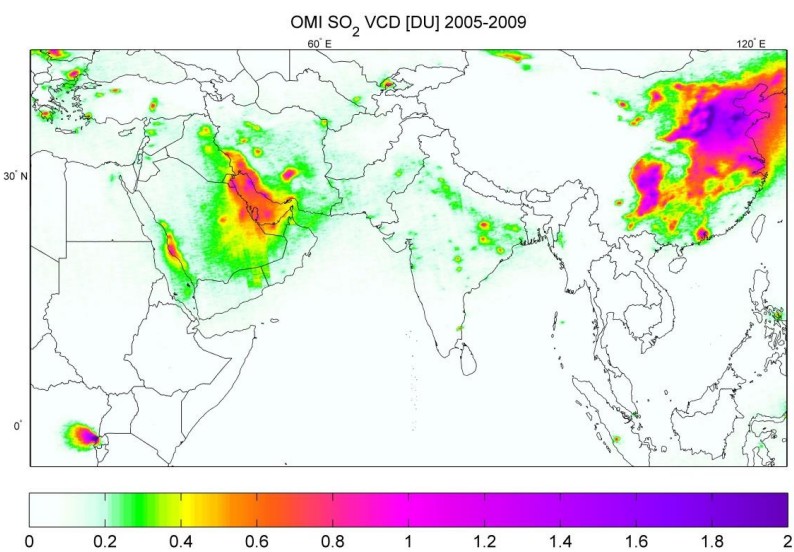

2    Figure 1: Map of averaged SO$_2$ columns from OMI clear-sky pixels for the 2005-2009 period.

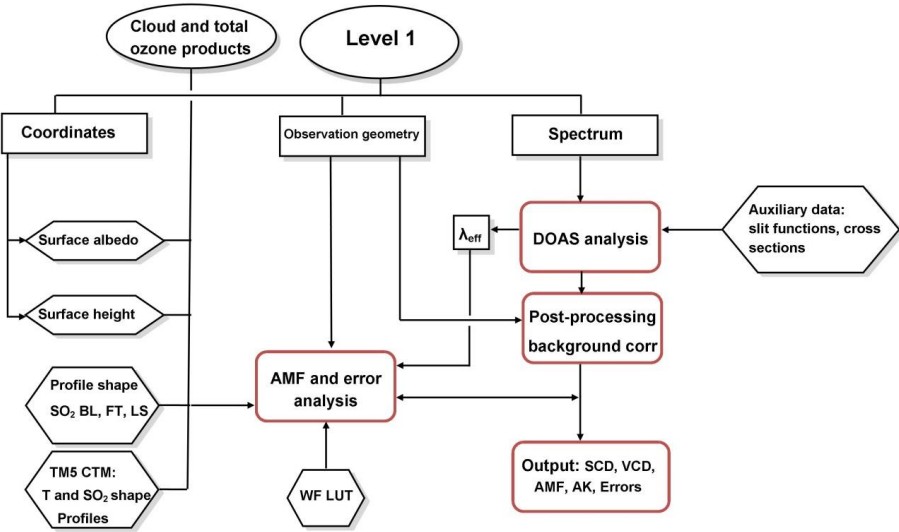

4    Figure 2. Flow Diagram of the TROPOMI DOAS retrieval algorithm for SO$_2$.



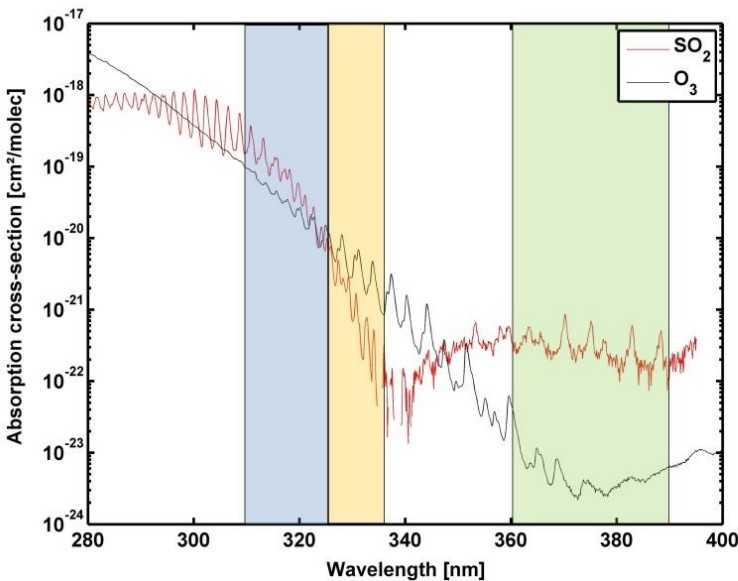

2    Figure 3. Absorption cross-sections of $SO_2$ and $O_3$. The blue, yellow and green boxes delimit

3    the three $SO_2$ fitting windows 312-326 nm, 325-335 nm and 360-390 nm, respectively.



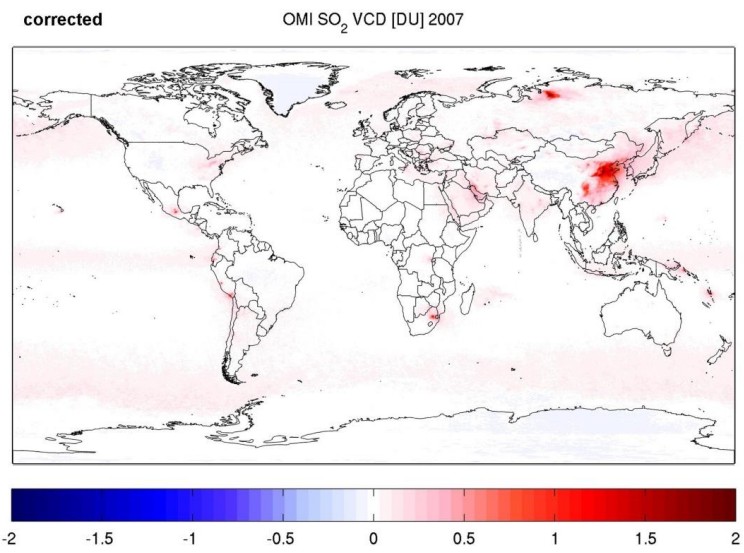

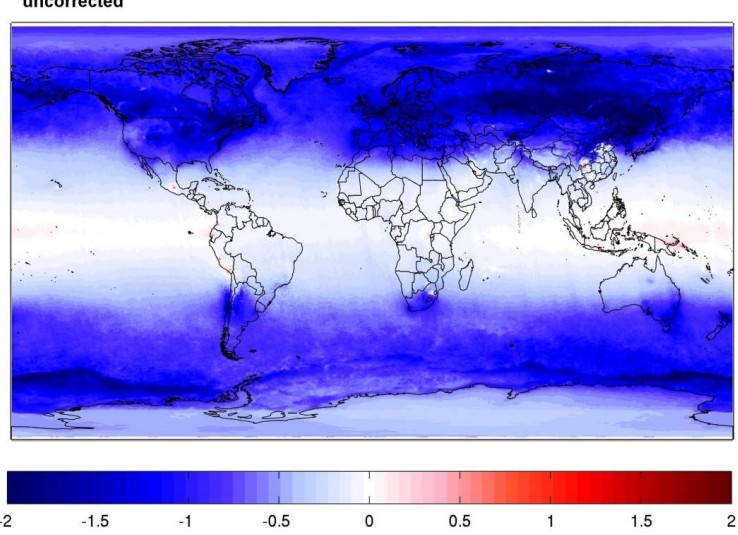

Figure 4. OMI SO₂ vertical columns (DU) averaged for the year 2007 (top) with and (bottom)
without background correction.   Only clear sky pixels (cloud fraction lower than 30%) have
been kept. AMFs calculated from SO₂ profiles from the IMAGES global model are applied to
the slant columns (Theys et al., 2015).





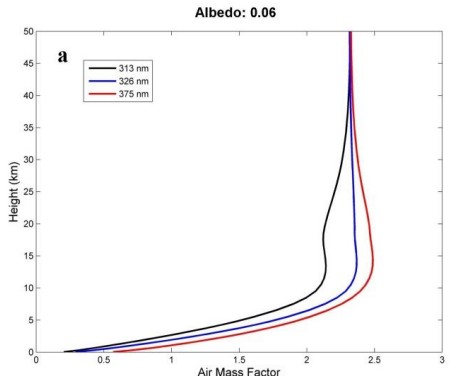 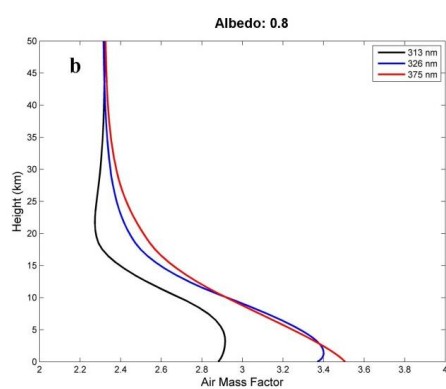

Figure 5. SO$_2$ box-AMFs at 313, 326 and 375nm for albedo of (a) 0.06 and (b) 0.8. SZA: 40°,
LOS: 10°, RAA: 0°, Surface height: 0 km.

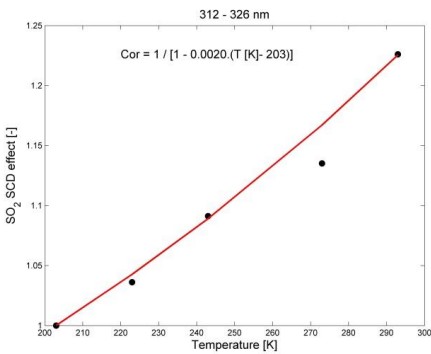 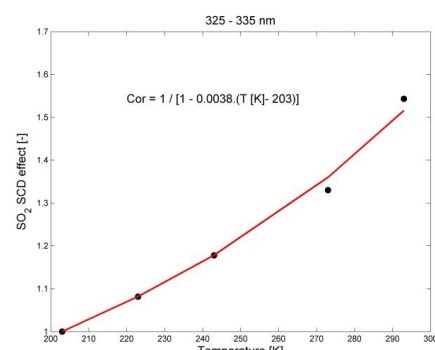

Figure 6. Effect of temperature (relative to 203K) on SO$_2$ retrieved SCD for fitting windows
312-326 nm (left) and 325-335 nm (right). The red lines show the adopted formulation of
$C_{temp}$ (Eq. 10). Note that, for the 312-326 nm window, the result at 273K has been discarded
from the fit as it is seems rather inconsistent with the dependence at other temperatures.




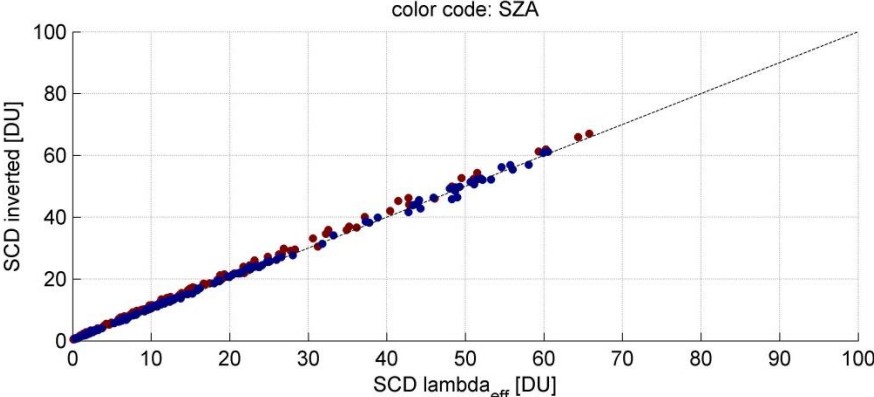

Figure 7. Retrieved SO$_2$ slant columns versus simulated SCDs at a wavelength of 313 nm from
synthetic spectra (SZA: 30°, 70°) in the spectral range 312-326 nm and for SO$_2$ layers in the
boundary layer, upper troposphere and lower stratosphere. The different points correspond
to different values for the line-of-sight angle (0, 45°), surface albedo (0.06, 0.8), surface
height (0, 5 km) and total ozone column (350, 500 DU).  SO$_2$ vertical columns as input of the
RT simulations are maximum of 25 DU.



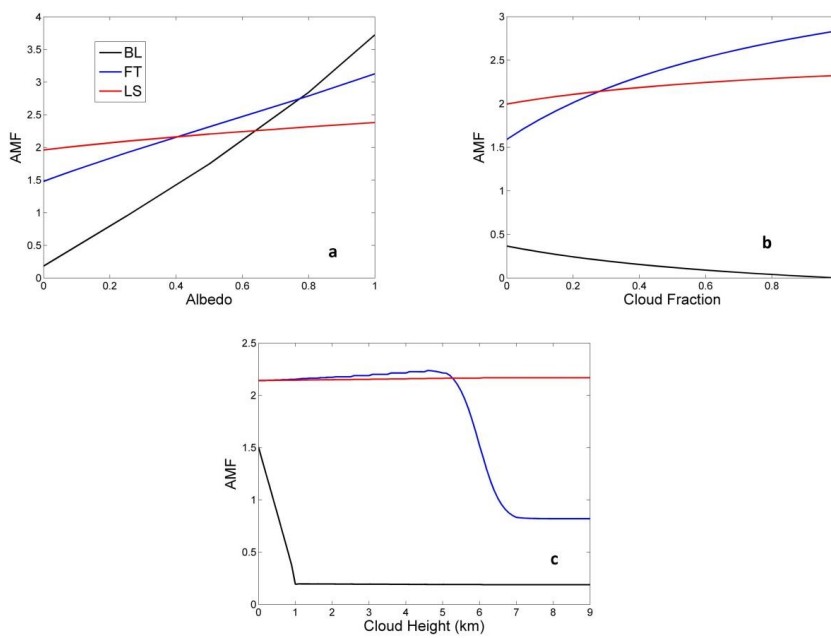

Figure 8. Air mass factors at 313 nm for $SO_2$ in the boundary layer (BL :0-1 km), free-
troposphere and lower stratosphere (FT, LS: Gaussian profiles with maximum height at 6,15
km and FWHM: 1 km). Calculations are for SZA=40°, Los=10°, RAA=0° and surface height=0
km. AMFs are displayed as a function of the (a) albedo for clear-sky conditions, (b) cloud
fraction for albedo=0.06, cloud albedo=0.8 and cloud top height=2km and (c) cloud top
height for albedo=0.06, cloud albedo=0.8 and cloud fraction=0.3.



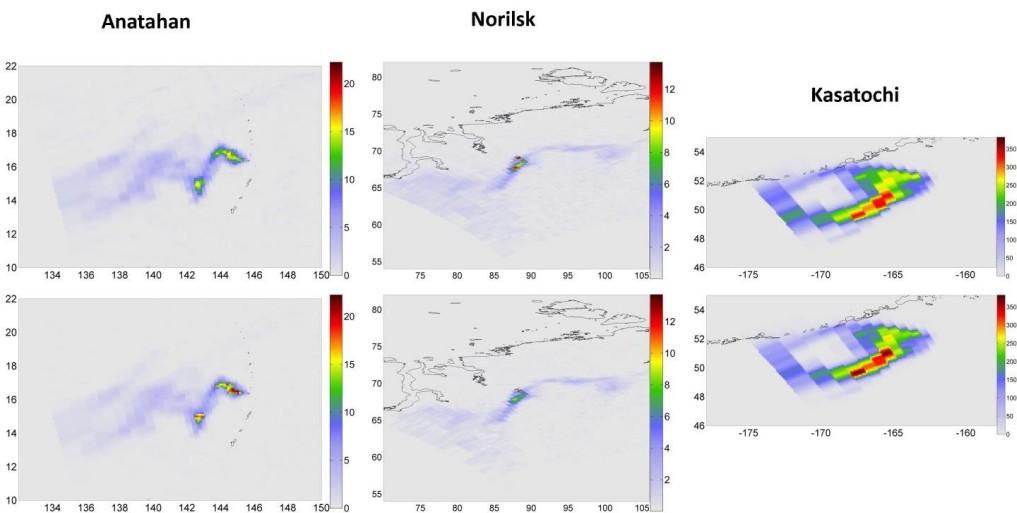

Figure 9. OMI SO$_2$ VCD (expressed in DU) for the Verification (upper panels) and Prototype
Algorithms (lower panels) for the three selected scenarios: during the Anatahan eruption
(left), over the Norilsk copper smelter area (center) and for the volcanic eruption of
Kasatochi (right). Note that, for each case, the colorbar has been scaled to the maximum SO$_2$
VCD from both algorithms.





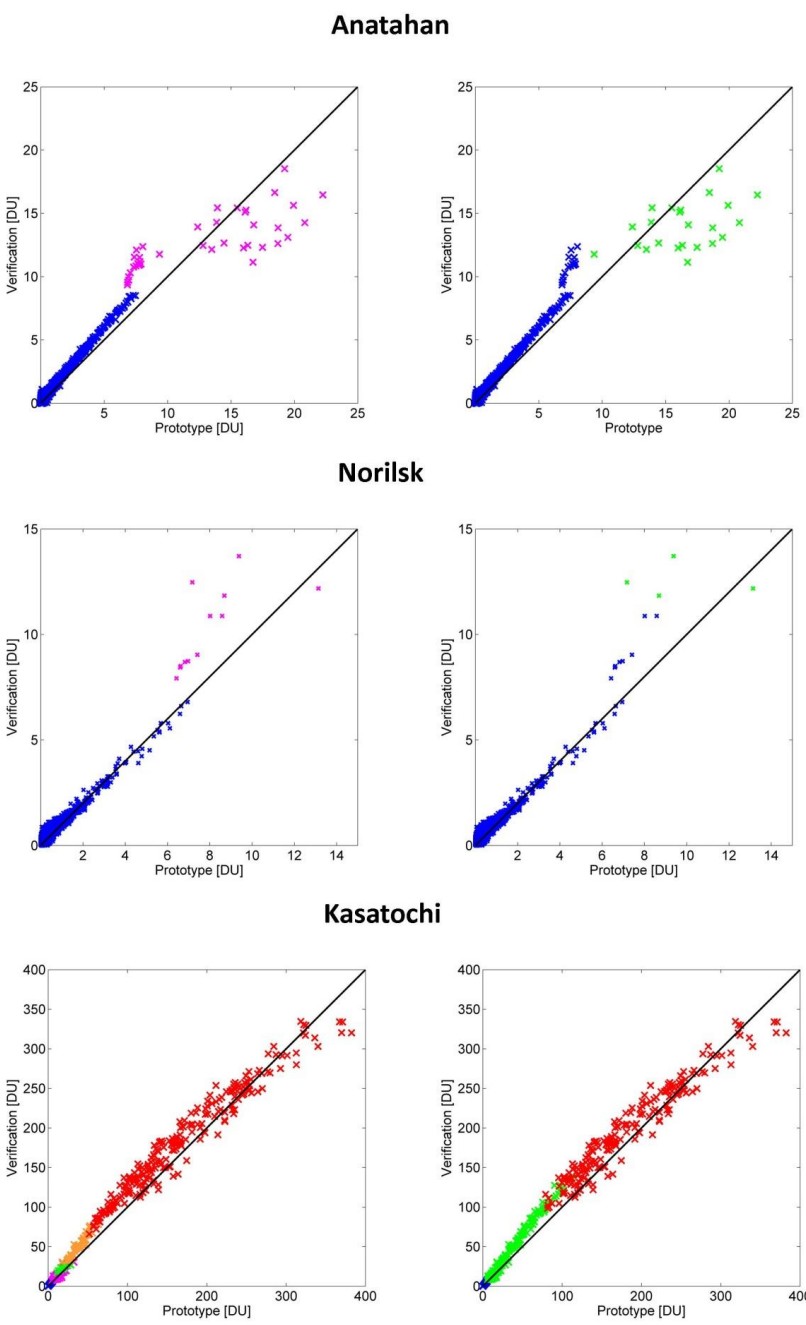

2    Figure 10. OMI SO$_2$ VCD (DU) scatter plots for PA (x-axis) and VA (y-axis) for the three test

3    cases, Anatahan eruption, Norislk anthropogenic emissions and Kasatochi eruption (from top





to bottom). The different fit windows used for both algorithms are color-coded: VA on left
panels (blue: SR, purple: SR/MR, green: MR, orange: MR/AR, red: AR), PA on right panels
(blue: 312-326 nm, green: 325-335 nm, red: 360-390 nm). For the three scenarios, the
prototype and verification algorithms agree fairly well with $r^2 \sim 0.9$.





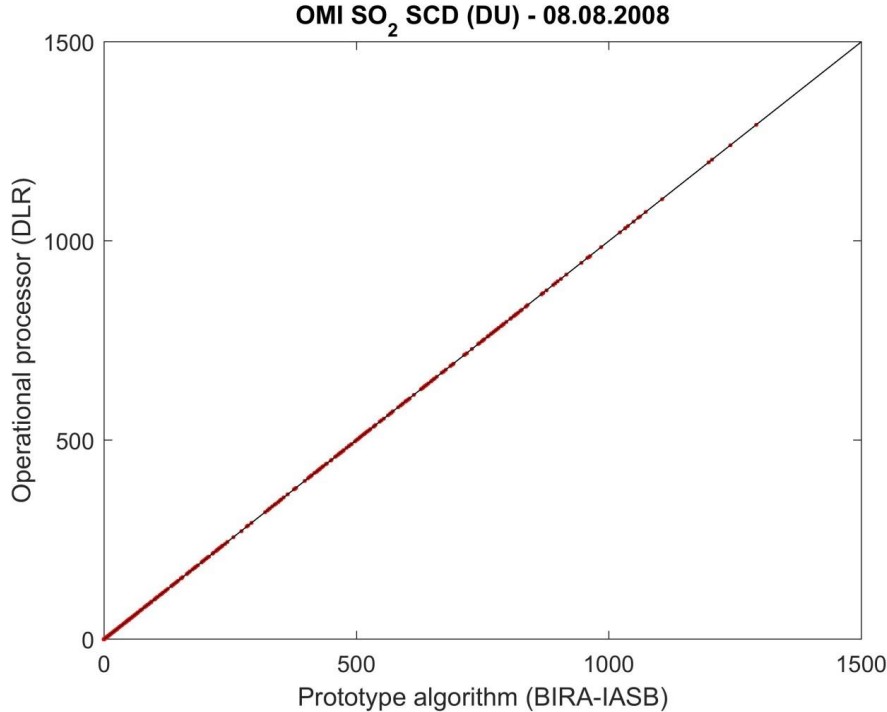

2   Figure 11. Comparison of SO$_2$ SCDs between prototype algorithm and operational processor

3   for the OMI test data of August 8, 2008.