# Peer review of "Sulfur dioxide retrievals from TROPOMI onboard Sentinel-5 Precursor: Algorithm Theoretical Basis"

_Atmospheric Measurement Techniques, 2016_

## Referee Comment (RC1) · Anonymous Referee #1 · 18 Oct 2016

This paper introduces the SO2 retrieval algorithm for the upcoming S-5P TROPOMI instrument. In addition to a description of the product requirement and retrieval algorithm (wavelength calibration, fitting window selection, DOAS analysis, bias correction, AMF calculation), the authors also provided a list of potential error sources in the retrievals and gave estimates of these errors where possible. Results from comparison with a verification algorithm (also based on the DOAS technique) were given, showing generally good agreement for a few volcanic cases, although the agreement between the two algorithms seem to be worse for higher SO2 loading. The authors also discussed future effort and potential data sources for product validation. Overall, this is a wellwritten and well-organized paper on an important product from the TROPOMI mission. The authors' effort in characterizing retrieval errors, while done in the absence of actual data before the launch of the instrument, is appreciated. I would recommend that the

paper be accepted for publication in AMT after some minor changes suggested below.

Page 2, line 21: another recent paper on long-term trend of SO2 is He et al., 2016:

He, H., Vinnikov, K. Y., Li, C., Krotkov, N. A., Jongeward, A. R., Li, Z., Stehr, J. W., Hains, J. C. and Dickerson, R. R. (2016), Response of SO2 and particulate air pollution to local and regional emission controls: A case study in Maryland. Earth's Future, 4: 94–109. doi:10.1002/2015EF000330.

Page 4, Line 6: consider spelling out the acronyms for the first use (e.g., CAPACITY and CAMELOT). A list of acronyms in the appendix may also help the readers.

Page 5, Line 23: special should be spatial?

Page 8, Line 12, maybe refer readers to Table A3 so that it is easier to find out which L2 cloud product will be used?

Page 9, Line 16: change "enables to define" to "enables us to define" or "enables defining".

Page 13, Line 27: "prior to the" DOAS "analysis"?

Page 14, Line 13: the average residual is calculated for the entire row or orbit? Also what is the percentage of pixels rejected in OMI?

Page 15, Line 22: typically how many pixels are available for averaging in each slant O3 column bin?

Page 20, Line 16: "what is the SO2 vertical profile" should be "what the SO2 vertical profile is".

Page 20, Line 24: so both fixed profiles and TM5 profiles will be used in the operational retrievals?

Page 21, Line 9: please specify how rescaling will be done.

Page 21, Line 12: if TM5 SO2 profile will be used, why not also use its temperature

AMTD
profile (instead of assuming a constant lapse rate everywhere).

Page 23, eq. 12: define m'

Page 24, Line 22: is it implied that a constant 0.2 DU positive bias exists in the slant column SO2?

Page 26, Line 28: how was this uncertainty estimated?

Page 27, Line 6: do you expect uncertainty in LIDORT in terms of comparison with VLIDORT or TOMRAD?

Page 31, Line 22: "error associated" should be "associated error".

Page 40, Line 6: "newly" should be "new"?

Page 42, Line 14: does the comparison between the two algorithms suggest that the measurement requirements will be met?

Table 6, #19: is 15% an upper limit for the effects of anthropogenic aerosols on air mass factor? Could this be an underestimate given the uncertainties in aerosol properties and vertical distribution?

Figure 11: I'm not sure that the figure is completely necessary: this is essentially the same program implemented in different systems and the results are expected to be very close if not identical.

AMTD

---

## Referee Comment (RC2) · Anonymous Referee #2 · 20 Oct 2016

- General Comments

This manuscript describes the SO2 retrieval algorithm of TROPOMI onboard Sentinel-5P. It well summarized the current status of the SO2 retrieval techniques, requirements, its error estimates, and validations. The retrieval strategy looks well-planned and this manuscript would be important for the future users of TROPOMI. Hence I would recommend publication of this manuscript for AMT after minor revisions below.

- Specific Comments

Page 18, lines 12-17 : The surface reflectance from Kleipool et al. (2008) generally provides reliable information, but there might be the better option for TROPOMI which has better spatial resolution (which is important for small urban and point source areas). At least discussion related to the spatial resolution of the surface reflectance and

alternatives for the database would be helpful for this moment of ATBD.

Page 25, lines 23-27 and Page 26, lines 11-14 : The measurement accuracy/uncertainty is major error source of SO2 retrieval with its measurement sensitivity as the author stated. Thus, it is valuable to add more details of the predicted measurement uncertainties. It might be challenging to evaluate/anticipate the measurement errors at this moment, but at least the authors could add brief discussion based on the requirements of L1b measurement and prior missions so that how much improvements would be expected. The radiometric calibration might include spectrally high-frequency errors such as the stray light and polarization sensitivity (particularly for window 1, from 312 to 326 nm). This might be included in Error source 8, but I would recommend to list those specifically since they are known high-frequency error sources.

- Technical corrections Page 17, line 16 and Page 19, line 8 : Please check if the Ac is defined in the manuscript. Please clarify the definition of fc and feff which are used for "effective cloud fraction".

Page 23, line 14 : I would recommend to use different symbol for error to distinguish from the absorption cross section.

---

## Author Comment (AC1) · 18 Nov 2016

Replies to the reviewers comments are below in italic; changes in the text are in blue in the revised version of the manuscript.

Reply: The authors gratefully thank all the reviewers for their thorough review and interesting comments which contributed to improve the manuscript. Review of anonymous Referee #1 This paper introduces the SO2 retrieval algorithm for the upcoming S-5P TROPOMI instrument. In addition to a description of the product requirement and retrieval algorithm (wavelength calibration, fitting window selection, DOAS analysis, bias correction, AMF calculation), the authors also provided a list of potential error sources in the retrievals and gave estimates of these errors where possible. Results from comparison with a verification algorithm (also based on the DOAS technique) were given,

showing generally good agreement for a few volcanic cases, although the agreement between the two algorithms seem to be worse for higher SO2 loading. The authors also discussed future effort and potential data sources for product validation. Overall, this is a well-written and well-organized paper on an important product from the TROPOMI mission. The authors' effort in characterizing retrieval errors, while done in the absence of actual data before the launch of the instrument, is appreciated. I would recommend that the paper be accepted for publication in AMT after some minor changes suggested below. Page 2, line 21: another recent paper on long-term trend of SO2 is He et al., 2016: He, H., Vinnikov, K. Y., Li, C., Krotkov, N. A., Jongeward, A. R., Li, Z., Stehr, J. W., Hains, J. C. and Dickerson, R. R. (2016), Response of SO2 and particulate air pollution to local and regional emission controls: A case study in Maryland. Earth's Future, 4: 94–109. doi:10.1002/2015EF000330.

Reply: changed in the text.

Page 4, Line 6: consider spelling out the acronyms for the first use (e.g., CAPACITY and CAMELOT). A list of acronyms in the appendix may also help the readers.

Reply: changed in the text. A list of acronyms has been added in Annex.

Page 5, Line 23: special should be spatial?

Reply: changed in the text.

Page 8, Line 12, maybe refer readers to Table A3 so that it is easier to find out which L2 cloud product will be used?

Reply: changed in the text.

Page 9, Line 16: change "enables to define" to "enables us to define" or "enables defining".

Reply: changed in the text.

Page 13, Line 27: "prior to the" DOAS "analysis"?

Reply: yes, changed in the text.

Page 14, Line 13: the average residual is calculated for the entire row or orbit? Also what is the percentage of pixels rejected in OMI?

Reply: we believe there is a misunderstanding. The word "pixel" refers to "detector pixel". The spike removal scheme applies on wavelength dimension not on across-track or along-track dimensions. For OMI, only few wavelengths are discarded from the DOAS analysis (maintaining largely the inverse problem as over-constrained) for OMI observations mostly in the South Atlantic region. We have clarified the text.

Page 15, Line 22: typically how many pixels are available for averaging in each slant O3 column bin?

Reply: not all O3 bins are populated equally but the number of pixels varies in the range of 3x10e3-3x10e4 per across-track position and for 2 weeks moving averages. For TROPOMI, we except double more pixels. We propose not to report those numbers in the text as they are not essential for the understanding of the method.

Page 20, Line 16: "what is the SO2 vertical profile" should be "what the SO2 vertical profile is".

Reply: changed in the text.

Page 20, Line 24: so both fixed profiles and TM5 profiles will be used in the operational retrievals?

Reply: yes, we have made it clearer in the text.

Page 21, Line 9: please specify how rescaling will be done.

Reply: we have added the following sentence: The TM5 SO2 profile is shifted to start at ps and scaled so that volume mixing ratios are preserved (see Zhou et al., 2009).

Page 21, Line 12: if TM5 SO2 profile will be used, why not also use its temperature

profile (instead of assuming a constant lapse rate everywhere).

Reply: it could be done but the impact on the AMF will be very small.

Page 23, eq. 12: define m'

Reply: m' is the weighting function. We have clarified the text.

Page 24, Line 22: is it implied that a constant 0.2 DU positive bias exists in the slant column SO2?

Reply: no, on average the bias is zero but in some places, SCDs might be biased (low or high) by max 0.2 DU (in absolute value).

Page 26, Line 28: how was this uncertainty estimated?

Reply: It has been estimated from averaged background corrected SCDs over clean regions (e.g. deserts). It has been clarified in the text.

Page 27, Line 6: do you expect uncertainty in LIDORT in terms of comparison with VLIDORT or TOMRAD?

Reply: We have not compared LIDORT with VLIDORT or TOMRAD but we expect relatively small error contribution from RTM (see also Error source # 12, section 3.2.2).

Page 31, Line 22: "error associated" should be "associated error".

Reply: changed in the text.

Page 40, Line 6: "newly" should be "new"?

Reply: changed in the text.

Page 42, Line 14: does the comparison between the two algorithms suggest that the measurement requirements will be met?

Reply: yes, this is true. It is also mentioned in Table 1 that requirements for volcanic SO2 will be met for VCD>0.5 DU. We have added a reference to section 4 (on verification results) in the section 2.1 on product requirements.

Table 6, #19: is 15% an upper limit for the effects of anthropogenic aerosols on air mass factor? Could this be an underestimate given the uncertainties in aerosol properties and vertical distribution?

Reply: we agree with the referee. Clearly the uncertainties due to aerosols can be much larger than 15% because of uncertainties on aerosol properties and vertical distribution (as mentioned on page 31). In table 6, we intended to simply report on published estimates but apparently there is a risk that the reader over-interprets the numbers and we propose to write 15% as an approximate value and not as an upper limit as it is now.

Figure 11: I'm not sure that the figure is completely necessary: this is essentially the same program implemented in different systems and the results are expected to be very close if not identical.

Reply: the SO2 prototype algorithm has not been provided to the operational team as a plug-and-play software package and the operation team had to rewrite most of the algorithmic modules. It required a lot of work and technical verification which is summarized in Fig 11. Although it is true that the figure is not essential, we think it also does not hurt to have a summarizing figure to illustrate the complete implementation of the algorithm in the operational environment.

Review of anonymous Referee #2

- General Comments

This manuscript describes the SO2 retrieval algorithm of TROPOMI onboard Sentinel-5P. It well summarized the current status of the SO2 retrieval techniques, requirements, its error estimates, and validations. The retrieval strategy looks well-planned and this manuscript would be important for the future users of TROPOMI. Hence I would recommend publication of this manuscript for AMT after minor revisions below.

- Specific Comments Page 18, lines 12-17 : The surface reflectance from Kleipool et al. (2008) generally provides reliable information, but there might be the better option for TROPOMI which has better spatial resolution (which is important for small urban and point source areas). At least discussion related to the spatial resolution of the surface reflectance and alternatives for the database would be helpful for this moment of ATBD. Reply: clearly, other surface reflectance databases more appropriate for S5P will become available and will be considered in next algorithmic versions. We have added a paragraph on this in the text. Page 25, lines 23-27 and Page 26, lines 11-14 : The measurement accuracy/uncertainty is major error source of SO2 retrieval with its measurement sensitivity as the author stated. Thus, it is valuable to add more details of the predicted measurement uncertainties. It might be challenging to evaluate/anticipate the measurement errors at this moment, but at least the authors could add brief discussion based on the requirements of L1b measurement and prior missions so that how much improvements would be expected. The radiometric calibration might include spectrally high-frequency errors such as the stray light and polarization sensitivity (particularly for window 1, from 312 to 326 nm). This might be included in Error source 8, but I would recommend to list those specifically since they are known high-frequency error sources.

Reply: We understand the concern of the reviewer and we agree that evaluating these measurement errors is challenging at the moment. Once the instrument will be operating, it will certainly be an important task to characterize instrumental-related spectral features, assess whether they interfere with SO2 retrievals and correct for those (either as part of L1b processor or in the form of pseudo absorption cross-sections in the DOAS analysis). We have added the following lines in Error source 8: Unknown or untreated instrumental characteristics such as stray light and polarization sensitivity can introduce spectral features that may lead to bias in the retrieved slant column data. To certain extend these can be prevented by the DOAS polynomial and the intensity offset correction settings, as long as the perturbing signals are a smooth function of wavelength. Conversely, high-frequency spectral structures can have potentially a

large impact on SO2 retrievals depending on their amplitude and whether they interfere with SO2 absorption structures. At the time of writing, it is hard to evaluate these measurement errors. Once the instrument will be operating, such measurement errors will be characterized and correct for, either as part of L1b processor or in the form of pseudo absorption cross-sections in the DOAS analysis.

- Technical corrections Page 17, line 16 and Page 19, line 8 : Please check if the Ac is defined in the manuscript. Please clarify the definition of fc and feff which are used for "effective cloud fraction".

Reply: We have defined Ac and clarify the text when needed.

Page 23, line 14 : I would recommend to use different symbol for error to distinguish from the absorption cross section.

Reply: we prefer to keep these symbols because they were adopted in other S5P ATBDs. Moreover, in this section, it is clear they are not referring to absorption cross-sections.

Please also note the supplement to this comment:
http://www.atmos-meas-tech-discuss.net/amt-2016-309/amt-2016-309-AC1-supplement.pdf

---

## Author Comment (AC2) · 18 Nov 2016

Replies to the reviewers comments are below in italic; changes in the text are in blue in the revised version of the manuscript. Reply: The authors gratefully thank all the reviewers for their thorough review and interesting comments which contributed to improve the manuscript. Review of anonymous Referee #1 This paper introduces the SO2 retrieval algorithm for the upcoming S-5P TROPOMI instrument. In addition to a description of the product requirement and retrieval algorithm (wavelength calibration, fitting window selection, DOAS analysis, bias correction, AMF calculation), the authors also provided a list of potential error sources in the retrievals and gave estimates of these errors where possible. Results from comparison with a verification algorithm (also based on the DOAS technique) were given, showing generally good agreement for a few volcanic cases, although the agreement between the two algorithms seem to be worse for higher SO2 loading. The authors also discussed future effort and potential data sources for product validation. Overall, this is a well-written and well-organized paper on an important product from the TROPOMI mission. The authors' effort in characterizing retrieval errors, while done in the absence of actual data before the launch of the instrument, is appreciated. I would recommend that the paper be accepted for publication in AMT after some minor changes suggested below. Page 2, line 21: another recent paper on long-term trend of SO2 is He et al., 2016: He, H., Vinnikov, K. Y., Li, C., Krotkov, N. A., Jongeward, A. R., Li, Z., Stehr, J. W., Hains, J. C. and Dickerson, R. R. (2016), Response of SO2 and particulate air pollution to local and regional emission controls: A case study in Maryland. Earth's Future, 4: 94–109. doi:10.1002/2015EF000330.

Reply: changed in the text.

Page 4, Line 6: consider spelling out the acronyms for the first use (e.g., CAPACITY and CAMELOT). A list of acronyms in the appendix may also help the readers.

Reply: changed in the text. A list of acronyms has been added in Annex.

Page 5, Line 23: special should be spatial?

Reply: changed in the text.

Page 8, Line 12, maybe refer readers to Table A3 so that it is easier to find out which L2 cloud product will be used?

Reply: changed in the text.

Page 9, Line 16: change "enables to define" to "enables us to define" or "enables defining".

Reply: changed in the text.

Page 13, Line 27: "prior to the" DOAS "analysis"?

Reply: yes, changed in the text.

Page 14, Line 13: the average residual is calculated for the entire row or orbit? Also what is the percentage of pixels rejected in OMI?

Reply: we believe there is a misunderstanding. The word "pixel" refers to "detector pixel". The spike removal scheme applies on wavelength dimension not on across-track or along-track dimensions. For OMI, only few wavelengths are discarded from the DOAS analysis (maintaining largely the inverse problem as over-constrained) for OMI observations mostly in the South Atlantic region. We have clarified the text.

Page 15, Line 22: typically how many pixels are available for averaging in each slant O3 column bin?

Reply: not all O3 bins are populated equally but the number of pixels varies in the range of 3x10e3-3x10e4 per across-track position and for 2 weeks moving averages. For TROPOMI, we except double more pixels. We propose not to report those numbers in the text as they are not essential for the understanding of the method.

Page 20, Line 16: "what is the SO2 vertical profile" should be "what the SO2 vertical profile is".

Reply: changed in the text.

Page 20, Line 24: so both fixed profiles and TM5 profiles will be used in the operational retrievals?

Reply: yes, we have made it clearer in the text.

Page 21, Line 9: please specify how rescaling will be done.

Reply: we have added the following sentence: The TM5 SO2 profile is shifted to start at ps and scaled so that volume mixing ratios are preserved (see Zhou et al., 2009).

Page 21, Line 12: if TM5 SO2 profile will be used, why not also use its temperature profile (instead of assuming a constant lapse rate everywhere).

Reply: it could be done but the impact on the AMF will be very small.

Page 23, eq. 12: define m'

Reply: m' is the weighting function. We have clarified the text.

Page 24, Line 22: is it implied that a constant 0.2 DU positive bias exists in the slant column SO2?

Reply: no, on average the bias is zero but in some places, SCDs might be biased (low or high) by max 0.2 DU (in absolute value).

Page 26, Line 28: how was this uncertainty estimated?

Reply: It has been estimated from averaged background corrected SCDs over clean regions (e.g. deserts). It has been clarified in the text.

Page 27, Line 6: do you expect uncertainty in LIDORT in terms of comparison with VLIDORT or TOMRAD?

Reply: We have not compared LIDORT with VLIDORT or TOMRAD but we expect relatively small error contribution from RTM (see also Error source # 12, section 3.2.2).

Page 31, Line 22: "error associated" should be "associated error".

Reply: changed in the text.

Page 40, Line 6: "newly" should be "new"?

Reply: changed in the text.

Page 42, Line 14: does the comparison between the two algorithms suggest that the measurement requirements will be met?

Reply: yes, this is true. It is also mentioned in Table 1 that requirements for volcanic SO2 will be met for VCD>0.5 DU. We have added a reference to section 4 (on verification results) in the section 2.1 on product requirements.

Table 6, #19: is 15% an upper limit for the effects of anthropogenic aerosols on air mass factor? Could this be an underestimate given the uncertainties in aerosol properties and vertical distribution?

Reply: we agree with the referee. Clearly the uncertainties due to aerosols can be much larger than 15% because of uncertainties on aerosol properties and vertical distribution (as mentioned on page 31). In table 6, we intended to simply report on published estimates but apparently there is a risk that the reader over-interprets the numbers and we propose to write 15% as an approximate value and not as an upper limit as it is now.

Figure 11: I'm not sure that the figure is completely necessary: this is essentially the same program implemented in different systems and the results are expected to be very close if not identical.

Reply: the SO2 prototype algorithm has not been provided to the operational team as a plug-and-play software package and the operation team had to rewrite most of the algorithmic modules. It required a lot of work and technical verification which is summarized in Fig 11. Although it is true that the figure is not essential, we think it also does not hurt to have a summarizing figure to illustrate the complete implementation of the algorithm in the operational environment.

Review of anonymous Referee #2

- General Comments

This manuscript describes the SO2 retrieval algorithm of TROPOMI onboard Sentinel-5P. It well summarized the current status of the SO2 retrieval techniques, requirements, its error estimates, and validations. The retrieval strategy looks well-planned and this manuscript would be important for the future users of TROPOMI. Hence I would recommend publication of this manuscript for AMT after minor revisions below.

- Specific Comments Page 18, lines 12-17 : The surface reflectance from Kleipool et al. (2008) generally provides reliable information, but there might be the better option for TROPOMI which has better spatial resolution (which is important for small urban and point source areas). At least discussion related to the spatial resolution of the surface reflectance and alternatives for the database would be helpful for this moment of ATBD. Reply: clearly, other surface reflectance databases more appropriate for S5P will become available and will be considered in next algorithmic versions. We have added a paragraph on this in the text. Page 25, lines 23-27 and Page 26, lines 11-14 : The measurement accuracy/uncertainty is major error source of SO2 retrieval with its measurement sensitivity as the author stated. Thus, it is valuable to add more details of the predicted measurement uncertainties. It might be challenging to evaluate/anticipate the measurement errors at this moment, but at least the authors could add brief discussion based on the requirements of L1b measurement and prior missions so that how much improvements would be expected. The radiometric calibration might include spectrally high-frequency errors such as the stray light and polarization sensitivity (particularly for window 1, from 312 to 326 nm). This might be included in Error source 8, but I would recommend to list those specifically since they are known high-frequency error sources.

Reply: We understand the concern of the reviewer and we agree that evaluating these measurement errors is challenging at the moment. Once the instrument will be operating, it will certainly be an important task to characterize instrumental-related spectral features, assess whether they interfere with SO2 retrievals and correct for those (either as part of L1b processor or in the form of pseudo absorption cross-sections in the DOAS analysis). We have added the following lines in Error source 8: Unknown or untreated instrumental characteristics such as stray light and polarization sensitivity can introduce spectral features that may lead to bias in the retrieved slant column data. To certain extend these can be prevented by the DOAS polynomial and the intensity offset correction settings, as long as the perturbing signals are a smooth function of wavelength. Conversely, high-frequency spectral structures can have potentially a large impact on SO2 retrievals depending on their amplitude and whether they interfere with SO2 absorption structures. At the time of writing, it is hard to evaluate these measurement errors. Once the instrument will be operating, such measurement errors will be characterized and correct for, either as part of L1b processor or in the form of pseudo absorption cross-sections in the DOAS analysis.

- Technical corrections Page 17, line 16 and Page 19, line 8 : Please check if the Ac is defined in the manuscript. Please clarify the definition of fc and feff which are used for "effective cloud fraction".

Reply: We have defined Ac and clarify the text when needed.

Page 23, line 14 : I would recommend to use different symbol for error to distinguish from the absorption cross section.

Reply: we prefer to keep these symbols because they were adopted in other S5P ATBDs. Moreover, in this section, it is clear they are not referring to absorption cross-sections.

Please also note the supplement to this comment:
http://www.atmos-meas-tech-discuss.net/amt-2016-309/amt-2016-309-AC2-supplement.pdf

**Supplement:**

[revised manuscript text omitted]

Unknown or untreated instrumental characteristics such as  stray light and polarization sensitivity can introduce spectral features that may lead to bias in the retrieved slant column data. To certain extend these can be prevented by the DOAS polynomial and the intensity offset correction settings, as long as the perturbing signals are a smooth function of wavelength. Conversely, high-frequency spectral structures can have potentially a large impact on $SO_2$ retrievals depending on their amplitude and whether they interfere with $SO_2$ absorption structures. At the time of writing, it is hard to evaluate these measurement errors. Once the instrument will be operating, such measurement errors will be characterized and correct for, either as part of L1b processor or in the form of pseudo absorption cross-sections in the DOAS analysis.

[revised manuscript text omitted]

He, H., Vinnikov, K. Y., Li, C., Krotkov, N. A., Jongeward, A. R., Li, Z., Stehr, J. W., Hains, J. C. and Dickerson, R. R.: Response of $SO_2$ and particulate air pollution to local and regional emission controls: A case study in Maryland. Earth's Future, 4: 94–109. doi:10.1002/2015EF000330, 2016.

[revised manuscript text omitted]

LOS: 10°, RAA: 0°, Surface height: 0 km.

[Figure]

Figure 6. Effect of temperature (relative to 203K) on SO$_2$ retrieved SCD for fitting windows

312-326 nm (left) and 325-335 nm (right). The red lines show the adopted formulation of

$C_{temp}$ (Eq. 10). Note that, for the 312-326 nm window, the result at 273K has been discarded from the fit as it is seems rather inconsistent with the dependence at other temperatures.

[Figure]

Figure 7. Retrieved $SO_2$ slant columns versus simulated SCDs at a wavelength of 313 nm from synthetic spectra (SZA: 30°, 70°) in the spectral range 312-326 nm and for $SO_2$ layers in the boundary layer, upper troposphere and lower stratosphere. The different points correspond to different values for the line-of-sight angle (0, 45°), surface albedo (0.06, 0.8), surface height (0, 5 km) and total ozone column (350, 500 DU).  $SO_2$ vertical columns as input of the

RT simulations are maximum of 25 DU.

[Figure]

Figure 8. Air mass factors at 313 nm for $SO_2$ in the boundary layer (BL :0-1 km), free-troposphere and lower stratosphere (FT, LS: Gaussian profiles with maximum height at 6,15 km and FWHM: 1 km). Calculations are for  SZA=40°, Los=10°, RAA=0° and surface height=0 km. AMFs are displayed as a function of the (a) albedo for clear-sky conditions, (b) cloud fraction for albedo=0.06, cloud albedo=0.8 and cloud top height=2km and (c) cloud top height for albedo=0.06, cloud albedo=0.8 and cloud fraction=0.3.

[Figure]

Figure 9. OMI SO$_2$ VCD (expressed in DU) for the Verification (upper panels) and Prototype

Algorithms (lower panels) for the three selected scenarios: during the Anatahan eruption (left), over the Norilsk copper smelter area (center) and for the volcanic eruption of

Kasatochi (right). Note that, for each case, the colorbar has been scaled to the maximum SO$_2$

VCD from both algorithms.

**Anatahan**

[Figure]

**Norilsk**

**Kasatochi**

Figure 10. OMI SO$_2$ VCD (DU) scatter plots for PA (x-axis) and VA (y-axis) for the three test cases, Anatahan eruption, Norislk anthropogenic emissions and Kasatochi eruption (from top to bottom). The different fit windows used for both algorithms are color-coded: VA on left panels (blue: SR, purple: SR/MR, green: MR, orange: MR/AR, red: AR), PA on right panels (blue: 312-326 nm, green: 325-335 nm, red: 360-390 nm). For the three scenarios, the prototype and verification algorithms agree fairly well with $r^2 \sim 0.9$.

[Figure]

Figure 11. Comparison of SO$_2$ SCDs between prototype algorithm and operational processor for the OMI test data of August 8, 2008.